# Unpacking DPO and PPO: Disentangling Best Practices for Learning from Preference Feedback

**Hamish Ivison**♣♠    **Yizhong Wang**♣♠    **Jiacheng Liu**♣♠
**Zeqiu Wu**♠    **Valentina Pyatkin**♣♠    **Nathan Lambert**♣
**Noah A. Smith**♣♠    **Yejin Choi**♣♠    **Hannaneh Hajishirzi**♣♠

♣Allen Institute for AI   ♠University of Washington
hamishiv@cs.washington.edu

## Abstract

Learning from preference feedback has emerged as an essential step for improving the generation quality and performance of modern language models (LMs). Despite its widespread use, the way preference-based learning is applied varies wildly, with differing data, learning algorithms, and evaluations used, making disentangling the impact of each aspect difficult. In this work, we identify four core aspects of preference-based learning: **preference data**, **learning algorithm**, **reward model**, and **policy training prompts**, systematically investigate the impact of these components on downstream model performance, and suggest a recipe for strong learning for preference feedback. Our findings indicate that all aspects are important for performance, with better preference data leading to the largest improvements, followed by the choice of learning algorithm, the use of improved reward models, and finally the use of additional unlabeled prompts for policy training. Notably, PPO outperforms DPO by up to 2.5% in math and 1.2% in general domains. High-quality preference data leads to improvements of up to 8% in instruction following and truthfulness. Despite significant gains of up to 5% in mathematical evaluation when scaling up reward models, we surprisingly observe marginal improvements in other categories.

We publicly release the code used for training[1] and evaluating[2] our models, along with the models and datasets themselves[3].

## 1   Introduction

Modern language models (LMs) commonly undergo a final stage of training, called *learning from preference feedback*,[4] before deployment to end-users. This stage of training has been applied to many prominent language models like ChatGPT [45], Llama 3 [35], and Claude [4], and has been shown to improve performance across a wide number of capabilities, including instruction following [22], code [28], math [54], and summarisation [67]. Despite the widespread use and potential of this learning paradigm, applications of preference-based learning vary wildly, both in the data and learning algorithm used. As such, it is unclear what aspects of learning from preferences matter most for

---

[1]https://github.com/hamishivi/EasyLM

[2]https://github.com/allenai/open-instruct

[3]https://huggingface.co/collections/allenai/tulu-v25-suite-66676520fd578080e126f618

[4]Sometimes this stage is called reinforcement learning from human feedback (RLHF). However, the human and reinforcement learning aspects are not always present, while the learning from preferences aspect is always present. We will use the terms 'learning from preferences' and 'preference-based learning' interchangeably.

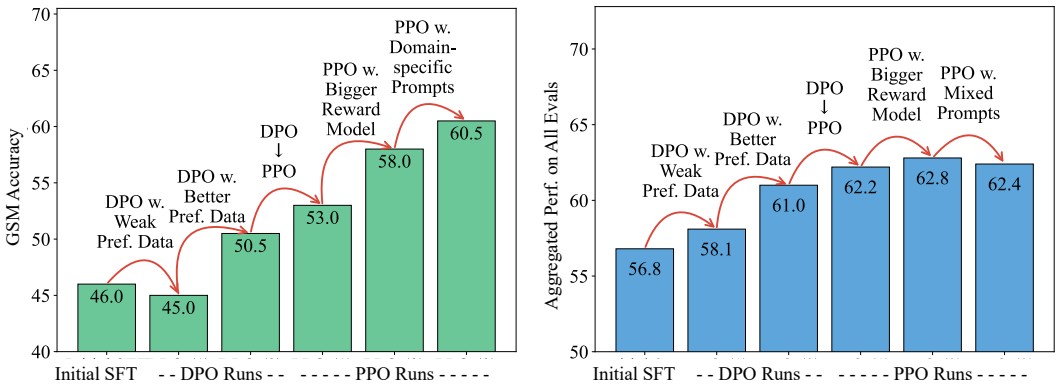

Figure 1: Performance improvements resulted by changing different components in the preference training of TÜLU. Left: Accuracy on GSM [9], for testing math capabilities. Right: Overall performance, aggregated over the 11 benchmarks described in §2.2.

downstream model performance, particularly in relation to the two most popular preference-based learning algorithms, PPO and DPO, which take different approaches to preference-based learning.

As seen in Figure 2, both DPO and PPO rely on training on **preference data**—DPO for directly training the model, and PPO for training a **reward model**. In PPO, this reward model is then used to score generations from the main model generated using a set of **policy training prompts** (unlabelled prompts used for eliciting generations). This provides us with four important aspects of learning from preferences, including the choice of **learning algorithm** itself (PPO vs. DPO).

In this work, we aim to systematically investigate these key components of learning from preferences, exploring the effect of varying each component on downstream model performance. Starting from an initial strong open supervised finetuning (SFT) model, we investigate each aspect in turn. As seen in Figure 1, we find that all aspects are important for performance, albeit to varying degrees. Our findings are as follows:

- When comparing 14 popular existing preference datasets across a diverse set of evaluations, we find that synthetic, diverse data annotated with per-aspect preferences works best for learning from preferences (§3.1). We find that the quality of the preferences (choice of chosen/rejected pairs) matters more than the quality of the actual generations being considered.
- PPO outperforms DPO across varied datasets in our evaluation suite, even when using exactly the same models and initial training data (§3.2).
- Increasing reward model size or dataset size used to train the reward model results in improved reward model performance *on benchmarks directly testing reward model performance*. Examining their effect on policy model performance when used during PPO training, we find that these improved reward models have a large effect on GSM performance, but give marginal to no improvements across all other evaluations considered (§3.3).
- Using unlabelled prompts that better match the test setting during policy can further improve model performance in domain-specific settings (e.g., when focusing on improving math performance), but has limited to no effect when targeting overall performance (§3.4).

Overall, we suggest a recipe for learning from preference feedback (§4): using synthetic preference datasets and training using PPO with a large reward model performs best overall. Additionally, targeted prompts should be used if one only cares about a single particular downstream task.

## 2   Setup

We first describe the core aspects of PPO and DPO before moving into our experimental setup. We summarise both approaches in Figure 2.

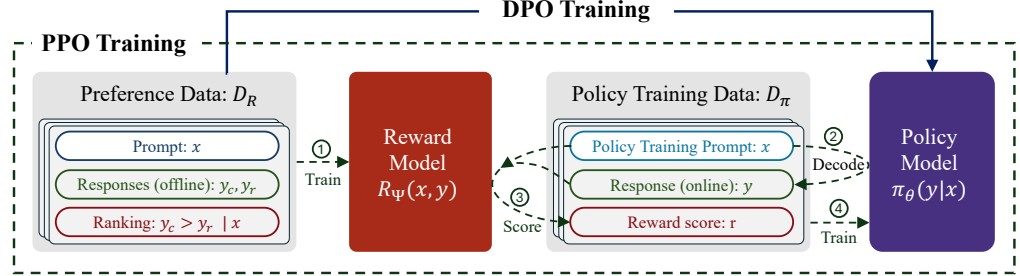

Figure 2: The core aspects of learning from preference feedback. For DPO (solid line), preference data is directly used to train a policy model. For PPO (dashed line), preference data is used to train a reward model, which is then used to score model-generated responses during PPO training.

## 2.1 PPO and DPO

**PPO.** PPO-based approaches for learning from preference feedback involve first training a reward model on preference data, and then training a policy model by scoring responses produced by the policy model itself during training ("online" learning) as shown in Fig. 2. First, the policy model is prompted to generate responses, which are then scored with the reward model. These scores (i.e., scalar rewards) are then used to guide the optimization of the policy model, following the PPO algorithm. Additionally, a KL penalty term is applied to avoid model degeneration.

- **Preference data.** A preference dataset $\mathcal{D}_R$ is used for training a reward model, and it typically consists of prompts, responses, and rankings. Each prompt $x$ comes with a pair of responses $y_c, y_r$, and a preference ranking between them (denoted as $y_c \succ y_r \mid x$, where $y_c$ is the chosen response and $y_r$ is the rejected one). Both the responses and the rankings can be either human-annotated, generated/evaluated by other language models, or derived from examining relative votes on publicly posted online comments (e.g., comparing high and low-upvoted comments from Reddit). While recent work has relied much on synthetic, model-generated data [53, 22], it is unclear if high-quality human data can provide similar or better performance, or if the significantly larger size of online-scraped datasets allows for improved performance.

- **Reward model.** The reward model $R_\psi(x, y)$ is a scalar function, and can be parameterized with a transformer where a regression head replaces the language modeling head. The reward model is usually trained to minimize the following loss:

$$\mathcal{L}_R(\psi) = -\mathbb{E}_{(x, y_c, y_r) \sim \mathcal{D}_R} \big[ \log \sigma \big( R_\psi(x, y_c) - R_\psi(x, y_r) \big) \big]. \tag{1}$$

- **Policy training prompts.** The policy training data $\mathcal{D}_\pi$ has a set of policy training prompts. Instead of the pre-generated responses, here for each prompt we sample a response $y$ from the policy model being actively trained, $y \sim \pi_\theta(y|x)$. It is then scored by the reward model to get a sequence-level reward, $r = R_\psi(x, y)$. Intuitively, this suggests that *more accurate rewards should improve model performance*. Additionally, we note that most existing works typically use either the prompts used during reward training [66] or a generic pool of anticipated user queries [37]. We investigate if better targeting the prompts used during policy training can further improve performance.

- **Policy training.** The goal of policy training is to maximize the reward of policy-generated responses, subject to a soft KL constraint that prevents degeneration:

$$\max_{\pi_\theta} \mathbb{E}_{x \sim \mathcal{D}_\pi, y \sim \pi_\theta(y|x)} \big[ R_\psi(x, y) \big] - \beta \mathbb{D}_{\mathrm{KL}} \big( \pi_\theta || \pi_{\mathrm{ref}} \big), \tag{2}$$

where $\pi_{\mathrm{ref}}$ is the reference policy (usually the same SFT policy that initializes policy training). We find tuning the KL penalty coefficient $\beta$ is important for performance and explore the effect of varying it in App. I. Since directly optimizing Eq. 2 can be unstable, it is common to reformulate language generation as a Markov decision process (MDP) and optimize using an RL algorithm. PPO is one of the most widely adopted RL algorithms for this problem. See App. F.2 for additional details about PPO.

**DPO.** DPO is an offline RL approach for performing learning from preference feedback. It allows one to directly optimize the policy on preference data, without building reward models or needing to sample online from the active policy. As such, DPO is an offline training algorithm. In simple terms, DPO aims at increasing the margin between the log-likelihood of the chosen responses and the log-likelihood of the rejected ones, while ensuring the model does not stray far from the initial policy.

- **Preference data.** The structure of preference data is identical to that of PPO.
- **Policy training.** By following a closed-form solution to Equation 2, DPO re-writes any reward model $R_\psi$ in terms of its corresponding optimal policy $\pi_{\theta*(\psi)}$, $R_\psi(x, y) = \beta \log \frac{\pi_{\theta*(\psi)}(y|x)}{\pi_{\text{ref}}(y|x)} + \beta \log Z(x)$, where $Z(x)$ is a partition function. Consequently, the policy model can be trained by directly optimizing the reward objective in Eq. 1, and hence the DPO loss:

$$\mathcal{L}_{\text{DPO}}(\theta) = -\mathbb{E}_{(x,y_c,y_r)\sim\mathcal{D}_R}\left[\log\sigma\left(\beta\log\frac{\pi_\theta(y_c\mid x)}{\pi_{\text{ref}}(y_c\mid x)} - \beta\log\frac{\pi_\theta(y_r\mid x)}{\pi_{\text{ref}}(y_r\mid x)}\right)\right]. \tag{3}$$

**Comparing DPO and PPO.** Compared with PPO, DPO is more efficient in terms of compute, speed, and engineering efforts. DPO does not need the extra stage of training a reward model, and during policy training it does not need to decode online responses (which is usually slow) or train an additional value model. Meanwhile, PPO trains on **online** data generated by the current policy, while DPO trains on static, pre-generated **offline** data. This may limit exploration in DPO and thus hurt the training quality. While concurrent work has compared DPO and PPO [60, 48], comparisons are typically limited to constrained domains and evaluations, using ground-truth rewards [60] or primarily examine smaller synthetic settings [48]. We complement such studies by comparing the downstream performance of models trained with DPO and PPO across a wide variety of datasets and evaluations and consider additional potential factors in PPO performance, such as improved reward models and policy training prompts.

## 2.2 Experimental and Evaluation Setup

We base our exploration off TÜLU 2 13B [22], a popular openly released model. TÜLU 2 is a series of Llama 2 [52] finetunes across all sizes with publicly released weights and data, the largest of which achieved state-of-the-art performance on AlpacaEval and Chatbot Arena. As such, we are curious how much further we can push the performance of TÜLU 2 models through exploring alternative datasets, training algorithms, reward models, and policy training prompts. We use this model as a base policy when training policy and reward models, following Ouyang et al. [38] for PPO and Rafailov et al. [39] for DPO. We provide additional details for each in App. F.

**Evaluation** We extend the TÜLU [55] evaluation, aiming to cover a diverse range of skills and behaviours. We evaluate models on 11 different benchmarks, covering skills such as factuality (MMLU [20]), reasoning (GSM8k [9], Big Bench Hard [5, 47]), truthfulness (TruthfulQA [29]), coding (HumanEval+ [6, 32], MBPP+ [2, 32]), safety (ToxiGen [19], XSTest [42]), and instruction following (AlpacaEval 1 and 2 [27, 13], IFEval [65]). We report the per-category average of evaluations and the overall average across categories. We provide further details in App. D.

## 3 Exploring Learning from Preference Feedback

We now explore each aspect of learning from preferences: **preference data**, **learning algorithm**, **reward models**, and finally **policy training prompts**.

### 3.1 Preference Data

We compare the performance of models trained with DPO across 14 different preference datasets in Table 1. We choose datasets that represent various potential sources of data: human-annotation (HH-RLHF [4], HelpSteer [56], Chatbot Arena 2023 [64] and 2024 [7], AlpacaFarm Human [14], PRM800k [28]), web-scraping (SHP-2 [15], StackExchange [25]), and synthetic generation (UltraFeedback [11], Nectar [66], Orca [34], Capybara [12], AlpacaFarm GPT-4 [14]). For UltraFeedback, we consider both using the 'overall' score provided in the dataset and taking an average of the per-aspect scores ('fine-grained'). We provide further detail on each dataset in App. B. We find that:

**Preference-based learning with existing datasets has the strongest effect on instruction following and truthfulness performance.** Our best models improve on the SFT model by over 8 points in these categories. In contrast, **preference-based learning does not aid factuality**, with all models remaining with 1 point of each other. This suggests that when using existing publically-available datasets, preference-based learning is most useful for improving chat-related abilities (instruction following, truthfulness) and learning stylistic features, but less strong at teaching new facts to a

| Source | | # Samples | Factuality | Reasoning | Coding | Truthfulness | Safety | Inst. Following | Average |
|---|---|---|---|---|---|---|---|---|---|
| - | Llama 2 base | - | 52.0 | 37.0 | 30.7 | 32.7 | 32.7 | - | - |
| - | TÜLU 2 (SFT) | - | 55.4 | 47.8 | 45.1 | 56.6 | 91.8 | 44.2 | 56.8 |
| Web | SHP-2 | 500,000 | 55.4 | 47.7 | 40.3 | 62.2 | 90.4 | 45.6 | 56.9 |
| | StackExchange | 500,000 | 55.7 | 46.8 | 39.6 | 67.4 | 92.6 | 44.6 | 57.8 |
| Human | PRM800k | 6,949 | 55.3 | 49.7 | **46.6** | 54.7 | 91.9 | 43.4 | 56.9 |
| | Chatbot Arena (2023) | 20,465 | 55.4 | 50.2 | 45.9 | 58.5 | 67.3 | 50.8 | 54.7 |
| | Chatbot Arena (2024) | 34,269 | 55.7 | 50.4 | 37.7 | 56.7 | 58.1 | 50.7 | 51.5 |
| | AlpacaF. Human Pref | 9,686 | 55.3 | 47.6 | 43.3 | 56.1 | 90.7 | 44.5 | 56.2 |
| | HH-RLHF | 158,530 | 54.7 | 46.0 | 43.6 | 65.6 | 93.1 | 45.4 | 58.1 |
| | HelpSteer | 9,270 | 55.2 | 48.2 | 46.5 | 60.3 | 92.5 | 45.2 | 58.0 |
| Synthetic | AlpacaF. GPT-4 Pref | 19,465 | 55.3 | 49.1 | 43.4 | 57.7 | 89.5 | 46.3 | 56.9 |
| | Capybara 7k | 7,563 | 55.2 | 46.4 | 46.4 | 57.5 | 91.5 | 46.1 | 57.2 |
| | Orca Pairs | 12,859 | 55.5 | 46.8 | 46.0 | 57.9 | 90.5 | 46.2 | 57.2 |
| | Nectar | 180,099 | 55.3 | 47.8 | 43.2 | 68.2 | 93.1 | 47.8 | 59.2 |
| | UltraF. (overall) | 60,908 | **55.6** | 48.8 | 46.5 | 67.6 | 92.1 | 51.1 | 60.3 |
| | UltraF. (fine-grained) | 60,908 | 55.3 | **50.9** | 45.9 | **69.3** | 91.9 | **52.8** | **61.0** |

Table 1: **Preference data:** Performance of TÜLU 2 13B models trained on various preference datasets using DPO. Blue indicates improvements over the SFT baseline, orange degradations. Overall, synthetic data works best. DPO training improves truthfulness and instruction-following most, with limited to no improvements in factuality and reasoning.

| Data / Model | Alg. | Factuality | Reasoning | Coding | Truthfulness | Safety | Inst. Foll. | Average |
|---|---|---|---|---|---|---|---|---|
| Llama 2 base | - | 52.0 | 37.0 | 30.7 | 32.7 | 32.7 | - | - |
| TÜLU 2 (SFT) | - | 55.4 | 47.8 | 45.1 | 56.6 | 91.8 | 44.2 | 56.8 |
| StackExchange | DPO | **55.3** | **47.8** | 42.4 | **56.2** | 92.0 | 46.7 | 56.7 |
| | PPO | 55.1 | **47.8** | **46.4** | 54.2 | **92.6** | **47.4** | **57.3** |
| ChatArena (2023) | DPO | **55.4** | **50.2** | 45.9 | **58.5** | 67.3 | **50.8** | 54.7 |
| | PPO | 55.2 | 49.2 | **46.4** | 55.8 | **79.4** | 49.7 | **55.9** |
| HH-RLHF | DPO | **55.2** | 47.6 | 44.2 | **60.0** | **93.4** | 46.6 | 57.8 |
| | PPO | 54.9 | **48.6** | **45.9** | 58.0 | 92.8 | **47.0** | **57.9** |
| Nectar | DPO | **55.6** | 45.8 | 39.0 | **68.1** | **93.3** | **48.4** | 58.4 |
| | PPO | 55.2 | **51.2** | **45.6** | 60.1 | 92.6 | 47.4 | **58.7** |
| UltraFeedback (FG) | DPO | 55.3 | 50.9 | 45.9 | 69.3 | **91.9** | 52.8 | 61.0 |
| | PPO | **56.0** | **52.0** | **47.7** | **71.5** | 91.8 | **54.4** | **62.2** |
| Avg. Δ b/w PPO & DPO | | -0.1 | +1.3 | +2.9 | -2.5 | +2.3 | +0.1 | +0.7 |

Table 2: **DPO vs PPO:** Average performance of 13B models trained using DPO and PPO across different datasets, along with the performance difference between DPO and PPO (Δ). Blue indicates improvements over the SFT baseline, orange degradations. All datasets are downsampled to 60,908 examples (except ChatArena, which is made up of 20,465 responses). PPO outperforms DPO by an average of 0.7 points, where most improvements are in reasoning, coding, and chat capabilities.

model. Interestingly, we observe that training on the Chatbot Arena data **performs poorly on safety**, suggesting Chatbot Arena volunteers generally prefer more toxic completions.

**Synthetic data with per-aspect annotations performs best.** Synthetic datasets generally outperform human-annotated and web-scraped datasets, especially in truthfulness. Additionally, using datasets collected by first getting per-aspect annotations (i.e., annotations from a human or model that score the helpfulness, harmlessness, etc. of the response independently) and then averaging across these scores tend to outperform datasets that rely only on overall judgements (i.e., just asking the annotator for an overall score instead of a per-aspect score). The two datasets that use this method, HelpSteer and UltraFeedback, display stronger or similar performance to datasets up to 15 times larger (e.g. HelpSteer vs HH-RLHF). We investigate the performance of varied sub-splits of UltraFeedback in App. E, which suggests that the use of per-aspect annotations is more important for performance than the quality of the models used to generate completions for the dataset.

| Reward Model | Direct Eval. | | PPO Training Perf. (w. UltraF. prompts) | | |
|---|---|---|---|---|---|
| | RewardBench Score | Best-of-N over SFT Avg. Perf. ($\Delta$) | GSM Acc. | AlpacaEval2 winrate | Avg. on All Evals. |
| 13B UltraF. RM | 61.0 | 56.9 (+5.8) | 53.0 | 26.1 | 62.2 |
| 13B Mix RM | **79.8** | 58.3 (+7.3) | 51.0 | 25.7 | 61.6 |
| 70B UltraF. RM | 73.6 | **61.1 (+10.3)** | **58.0** | 26.7 | **62.8** |
| 70B Mix RM | 73.9 | 60.6 (+9.5) | 51.5 | **31.6** | 61.8 |

Table 3: **Reward model evaluation:** (a) **Direct evaluation:** reward models when directly evaluated using RewardBench [26] and Best-of-N (left two columns); for BoN, we report both raw average performance, and the difference in performance over the base SFT model in brackets. (b) **Downstream evaluation:** models trained using PPO and the given reward model (right three columns). We report GSM and AlpacaEval 2 performance along with average performance across the entire evaluation suite defined in §2.2. Directly comparing reward models indicate increasing scale and data improves reward models, but these only minimally impact downstream performance.

## 3.2 Preference Learning Algorithm: DPO vs. PPO

We now compare algorithms for learning from preferences, comparing the performance of DPO and PPO when the same base models and data are used (Table 2). We use **exactly the same data** for training DPO and PPO models,[5] and subsample larger datasets to 60,908 examples due to computational resources and only use these subsampled datasets during reward model and PPO training. For dataset choice, we use the top-performing dataset from each source type in Table 1 (StackExchange from Web, HH-RLHF from human, Ultrafeedback from synthetic). We also include the second-best performing dataset overall (Nectar) and an additional human-annotated dataset from a popular evaluation platform (Chatbot Arena 2023). Results in Table 2 indicate that:

**PPO outperforms DPO.** Across all datasets, models trained with PPO outperform models trained with DPO. In fact, PPO is able to provide improvements over the SFT model in cases where DPO training did not, such as when using StackExchange. On average, PPO significantly[6] improves over DPO performance.

**PPO improves on DPO in reasoning, coding and safety capabilities most.** PPO improves over DPO by an average of 1.3, 2.9, and 2.3 points for reasoning, coding, and safety respectively, while truthfulness tends to degrade by an average of 2.5 points. Instruction following and factuality remain largely the same. Interestingly, we find that models trained with PPO are far more likely than DPO-trained models to perform chain-of-thought reasoning when prompted with reasoning or math problems, even when not given in-context examples using chain-of-thought. This suggests that reasoning improvements from PPO may be due to increased chain-of-thought abilities. Additionally, while overall instruction following ability remains similar, we find that PPO-trained models tend to perform better at AlpacaEval 1 and 2, with PPO-trained models outperforming DPO-trained ones on AlpacaEval 2 by an average of 3.4 points.

## 3.3 Reward Models

Next, we focus on PPO and study reward models both directly, and on downstream tasks with PPO training (Table 3). We first consider two ways to improve a reward model:

1. **Scaling up the training data for the reward model.** We construct a data mixture of the top-performing datasets in Table 1 from each section: UltraFeedback, HelpSteer, Nectar, StackExchange, HH-RLHF, and additionally add PRM800k for math data. We compare reward models trained on this data mixture (called **Mix RM**) with reward models trained only on UltraFeedback (**UltraF. RM**) – the top-performing dataset from prior sections.
2. **Scaling up the reward model size.** We train reward models at two scales of 13B and 70B starting from TÜLU 2.

---

[5]That is, we compare DPO models trained directly on each dataset with PPO models trained using reward models trained directly on that dataset and using prompts from the same dataset. Note that PPO uses additional generations from the model during training, but both approaches use the same amount of **labelled** data.

[6]$p < 0.05$ in a two-tailed paired t-test.

**Direct evaluation of reward models.** To isolate the performance of the differing reward models, we first evaluate them with best-of-N (BoN): we sample 16 generations from TÜLU 2 13B SFT, score them using the given reward model, and then use the top-scoring output as the model output. Notably, we ensure model outputs are identical between runs, meaning that the only difference is the reward model scores. We perform evaluation on the subset of evaluations in our suite that require long-form generation,[7] and report overall average performance. We additionally evaluate our reward models on RewardBench [26], a popular evaluation for reward models, which involves evaluating if the relative scores given by reward models match a test set of chosen-rejected pairs from diverse sources. We provide further details in Appendix H.

Results in Table 3 indicate that **either increasing the reward model dataset ('Mix') or reward model size (from 13B to 70B) improves direct RM performance**. Surprisingly, we find that our 70B reward models perform best on BoN evaluation, while the 13B mixture model performs best on RewardBench. Both evaluations show that increasing the dataset mixture and increasing the dataset size can help, although increasing the dataset mixture is not as useful for further improving the 70B reward model. Although scaling model size helps with best-of-N, it does not improve RewardBench performance. Examining the per-split performance on RewardBench in App. H Table 13, the largest gaps between the 13B Mix RM and the 70B Mix RM is in reasoning (mostly code), suggesting that the larger model may not benefit much from the additional (somewhat noisy) data[8].

**Downstream evaluation of reward models.** We then test if our improved reward models lead to improved downstream policy models when used during PPO training. We perform PPO training using the UltraFeedback prompts during training and report our results on the right side of Table 3. Surprisingly, we find that **improvements in reward models result in surprisingly small improvements in overall downstream performance**. We see the largest (and only) overall improvement when using the 70B UltraFeedback RM, despite the fact that all improved RMs performed significantly better than the 13B UltraFeedback RM in RewardBench and Best-of-N evaluations. Additionally, the improvement from the 70B RM is largely driven by a large performance jump in GSM, as shown in Table 3, while other metrics stay largely similar. This suggests that it is difficult to translate improvements in reward models to the underlying policy. We note that most prior work examining reward models tends to examine either direct reward model performance [26, 61] or proxy reward [17, 40], rather than downstream performance. **Our findings suggest that improving such evaluations may not necessarily translate to improved downstream performance.** Additionally, we find that using the larger 70B UltraF. RM is less sensitive and performs well with lower KL penalty coefficient values than using the 13B UltraF. RM, and further examine the effect of the KL penalty coefficient on performance in App. I.

While it may seem unintuitive that an extreme increase in reward model size does not lead to extreme improvements in performance, we note that prior work has similarly noted that much larger reward models do not necessarily lead to significant improvements in performance, and smaller reward models can lead to similar performance (Ouyang et al. [37] §C.2, Wang et al. [56] §4.3), although we are the first, to our knowledge, to ablate this and explicitly report results on downstream evaluations.

We additionally explore how further training and potentially overoptimizing against the reward model affects performance across different evaluations in App. L. Importantly, we find that different evaluations show different trends - while some evaluations such as AlpacaEval benefit from continued training, other evaluations such as IFEval or GSM8k drop with continued training. This highlights the importance of evaluating over a diverse set of test tasks, including both 'traditional' benchmarks and LLM-as-a-judge evaluations.

### 3.4 Policy Training Prompts

We finally examine the effect of using varied policy training prompts, first when targeting a particular capability (math performance as evaluated by GSM), and then for improving overall performance. This is in contrast to prior work that just re-use the prompts used for reward model training [66] or a generic pool of anticipated user queries [37].

---

[7]This is because best-of-N with multiple-choice evaluations does not test the ability of the reward model to pick model generations, but merely picks the correct multiple choice answer.

[8]As the RewardBench reasoning subset is made up mainly of coding tasks, we hypothesise the presence of noisy StackExchange data may harm the 70B model, which has more capacity to fit to the larger mixture dataset than the 13B RM.

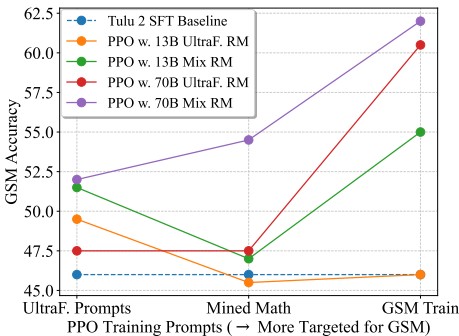

Figure 3: **Policy training prompt math evaluation:** Performance of models trained on 20K prompts from varying sources using PPO and evaluated on GSM. Training with larger RMs trained on more data benefits more from in-domain prompts (i.e., prompts directly from the GSM train set), while weaker RMs struggle to generalize beyond their training prompts.

| Reward Model | Prompts | GSM % | Coding | Avg. Across All Evals |
|---|---|---|---|---|
| Tulu 2 SFT | - | 46.0 | 45.1 | 56.8 |
| 13B UltraF. | UF | 53.0 | 47.7 | **62.2** |
| 13B UltraF. | Mixed | **54.5** | **47.8** | 61.9 |
| 13B Mix | UF | **51.0** | **46.8** | **61.6** |
| 13B Mix | Mixed | 50.5 | 43.8 | 60.9 |
| 70B UltraF. | UF | **58.0** | 47.3 | **62.8** |
| 70B UltraF. | Mixed | 56.5 | **48.4** | 62.4 |
| 70B Mix | UF | 51.5 | **46.1** | **61.8** |
| 70B Mix | Mixed | **52.0** | 44.9 | 61.1 |

Table 4: **Policy training prompt overall evaluation:** Performance of PPO policy models trained with the given reward models on 60K prompts from either UltraFeedback or the remixed prompt set that adds additional unlabeled math and coding-related prompts. Using the remixed prompt set does not improve performance, either on specific evaluations (math, code) or in terms of overall performance.

### 3.4.1 The effect of targeted prompts.

We first examine the effect of using policy training prompts targeted towards a particular downstream setting—specifically, math capabilities as evaluated by GSM. We construct three different prompt sets: prompts sourced from the GSM train set, math-related prompts from varied datasets, and 20K random prompts from UltraFeedback[9]. We further detail the approach used to identify math-related prompts in Appendix J. We then train models using PPO with each reward model from the previous section on each prompt set. Results in Fig. 3 demonstrate that:

**Larger reward models perform better when closely matching train prompts to test settings.** When using prompts from the GSM train set, we find using either 70B reward models during training leads to significant improvements in GSM, with our best performing model improving over the SFT model by 16 points (46%→62%).

**Weaker reward models struggle to make use of prompts differing from those they are trained on.** Surprisingly, we find that training with the 13B UltraFeedback reward model actually performs *worse* when using prompts apart from UltraFeedback, potentially due to the reward model generalising poorly to prompts not seen during training. Similarly, the 13B Mix and 70B UltraFeedback reward models struggle to make use of math prompts, which are out-of-domain for the reward models, but in-domain for the task.

Overall, these results suggest that **targeted prompt distributions can improve performance when coupled with powerful reward models** (e.g., using prompts from the GSM train set and then evaluating on GSM). This highlights a strength of PPO: **it can make use of unlabelled prompts to better target downstream tasks**.

### 3.4.2 Altering prompts to improve overall performance.

Inspired by the success of the targeted prompts, we construct a new remixed prompt set by finding 20K math and 20K code-related prompts using the same method as in the previous subsection (see App. J). We then combine the math, code, and UltraFeedback prompts to create a larger prompt pool that we downsample randomly to 60,908 prompts. This rebalances the prompt pool to focus more on coding and math tasks, which we hope yields improvements on math and code respectively while maintaining overall performance. We present our results in Table 4.

Surprisingly, we observe that **using mixed prompts does not seem to improve performance in the generalist setting**. When looking at code and math results specifically, we do not see consistent

---

[9]Note that this is only a third of all UltraFeedback data. We reduce the size to fairly compare to the small number of GSM8k prompts.

| Model | Factuality | Reasoning | Coding | Truthfulness | Safety | Instr. Foll. | Average |
|---|---|---|---|---|---|---|---|
| Llama 2 13B Base | 52.0 | 37.0 | 30.7 | 32.7 | 32.7 | - | - |
| Llama 2 Chat 13B [52] | 53.2 | 24.7 | 36.9 | **88.0** | 91.9 | 51.2 | 57.7 |
| Nous Hermes 13B [51] | 53.2 | 43.5 | 47.7 | 80.5 | 43.9 | 38.7 | 51.3 |
| Vicuna 1.5 13B [64] | 54.5 | 39.3 | 38.5 | 62.8 | 92.4 | 45.8 | 55.6 |
| TÜLU 2 13B SFT | 55.4 | 47.8 | 45.1 | 56.6 | 91.8 | 44.2 | 56.8 |
| TÜLU 2+DPO 13B | 55.3 | 50.9 | 45.9 | 69.3 | 91.9 | 52.8 | 61.0 |
| TÜLU 2+PPO 13B (13B UFRM) | 56.0 | 52.0 | 47.7 | 71.5 | 91.8 | 54.4 | 62.2 |
| TÜLU 2+PPO 13B (70B UFRM) | 55.4 | 53.9 | 47.3 | 72.3 | 91.9 | 55.8 | 62.8 |
| TÜLU 2+PPO 13B (70B UFRM+MP) | 55.3 | 53.1 | 48.4 | 71.0 | **92.7** | 54.0 | 62.4 |
| L3+TÜLU 2 8B SFT | 58.0 | 58.6 | 56.4 | 59.2 | 92.8 | 42.6 | 61.3 |
| L3+TÜLU 2+DPO 8B | 59.4 | 56.2 | 55.6 | 71.4 | 91.7 | 50.4 | 64.1 |
| L3+TÜLU 2+PPO 8B (8B UFRM) | **59.5** | 57.0 | **55.9** | 69.6 | 91.4 | **56.0** | 64.9 |
| L3+TÜLU 2+PPO 8B (70B UFRM) | 58.5 | **60.8** | 55.0 | 72.8 | 91.8 | 55.8 | **65.8** |
| L3+TÜLU 2+PPO 8B (70B UFRM+MP) | 58.3 | 40.6 | 48.2 | 62.4 | 91.0 | 53.0 | 58.9 |

Table 5: **Putting together a recipe for preference-based learning:** Performance of our best-performing models along with popular open models based on Llama 2 13B. 'MP' refers to using the mixed prompt set described in §4. 'L3' stands for experiments using Llama 3 as a base model. Using PPO with a large reward model performs best overall.

improvements using the altered prompt mixture, and our overall performance tends to drop. This is likely due to the already diverse nature of UltraFeedback, such that when looking at the whole dataset (i.e., not just the 20K subset in Figure 3), we are able to reach strong performance on math and coding evaluations. Altering the distribution away from other tasks slightly hurts the overall performance. We additionally found that training all the mined prompts in additional to all of UltraFeedback (i.e., not downsampling the combined prompt set) did not yield further improvements over the results shown in Table 4.

## 4 A Recipe for Learning from Preferences

Putting together all our findings from previous sections, we suggest a recipe for training a strong model using learning from preferences, as shown in Figure 1 and in Table 5. We take a **high-quality, synthetic preference dataset**, a **large reward model**,[10] and train it using **PPO**. If we additionally wish to focus on a specific domain, we can additionally collect **domain-specific prompts for policy training**. We find that our best model (TÜLU 2+PPO 13B trained with the 70B UltraF. RM) outperforms our best DPO model and other popular models based off Llama 2 13B, including Llama 2 Chat, which has also undergone SFT and PPO training. Additionally, incorporating task-specific prompts into policy training may further improve performance when the prompts align closely with downstream evaluations, as shown in Figure 3. Finally, we also experiment with Llama 3.0 8B [35], finetuning on the TÜLU2 Mix, and then training it using DPO and PPO with the same hyperparameters. We find that overall performance is significantly improved, and we observe similar trends as with our other experiments (DPO performing better than PPO, using a larger reward model improving performance, using mixed prompts not improving performance).

## 5 Related Work

**Learning from Preferences for LMs.**

Initial approaches to learning from preferences used reinforcement learning from human feedback (RLHF) [8, 67], a method that first trains a reward model to capture human preferences and then optimizes against it using reinforcement learning algorithms such as PPO [44]. Recent work has additionally questioned the need for PPO, and has found that similar but simpler approaches such as REINFORCE [46] with LM-specific adjustments work well [1, 50]. Unlike prior work, we instead focus on examining the effect of varying the **data and models** used in PPO (i.e., the reward model, preference data, initial policy model, and prompts used for sampling outputs). We believe that our results should transfer to similar approaches such as REINFORCE, since they share many

---

[10]However, a smaller 13B RM performs almost as well if one is compute-constrained.

commonalities with PPO (e.g., reliance on a well-tuned reward model and an unlabelled prompt set for eliciting generations during training).

Another line of recent work has also attempted to simplify the PPO algorithm and remove the online generation component, with the most popular algorithm following this approach being DPO [39]. DPO's ease of implementation and use has made it widely popular among open-source community. Notably, several high-performing models have been trained using DPO for learning from preferences, including TÜLU 2 [22], Zephyr [53], and Mixtral-Instruct [24]. Much recent work [3, 21, 59, 16, *inter alia*] has attempted to further improve the DPO algorithm. However, comparisons of these approaches so far have found that they largely perform similarly [41, 43]. As such, in this work we focus on the most popular variant, DPO, and examine what data works best for it and how it compares to a popular online RL approach, PPO.

**Comparing Approaches for Learning from Preferences.** Recent concurrent work has compared the properties and performance of DPO, PPO, and other approaches for learning from preference feedback. Xu et al. [60] suggest DPO performs poorly when using data out-of-distribution from the initial base model, while PPO (and a semi-online DPO variant) perform better both in such cases and overall when evaluated on safety and code capabilities. However, they do not investigate the impact of reward models and focus on core algorithmic details of PPO that lead to improvements. Tajwar et al. [48] identify on-policy sampling and negative gradients as two important aspects of preference-based learning when optimal reward values do not lie 'close' to the base model's distribution and the preference data is skewed. Tang et al. [49] study how the IPO [3] algorithm performs in the static offline setting versus various ways of updating or ordering the data in an online manner. In this work, we focus on empirically examining the impact of core aspects of learning from preference feedback, including the effects of varied rewards and data.

# 6 Conclusion

In this work, we have systematically explored the core components of learning from preference feedback and examined the relative impacts of each in turn on model performance across a wide range of evaluations. Our results suggest the following ordering of importance: preference data quality, algorithm choice, reward model quality, and finally targeted policy training prompts. Additionally, we find that using larger reward models can significantly improve math capabilities, but have marginal effects on other capabilities we evaluate in this work. Overall, we suggest a recipe for learning from preference feedback with currently available resources: best performance can be achieved by using a strong synthetic dataset (UltraFeedback), and using PPO training with a large reward model. Our work suggests that further exploring how to make better use of improved reward models is an important direction for further improving PPO-style approaches to learning from preference data. We plan to release models and code related to this paper and hope that our settings provide a firm base for future work further exploring learning from preferences for language models.

# Acknowledgements

This research was funded in part with funding from the Defense Advanced Research Projects Agency (DARPA) under Contract No. FA8650-23-C-7316, DARPA MCS program through NIWC Pacific (N66001-19-2-4031), and NSF IIS-2044660. Research supported with Cloud TPUs from Google's TPU Research Cloud (TRC). We thank members of AI2 and UW NLP for useful feedback throughout this project.

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

# A  Limitations & Broader Impacts

**Limitations.** As an empirical study with limited compute, our results are largely based on an in-depth examination over a single model suite (TÜLU 2), using two base models (Llama 2 and 3). Additionally, our evaluation may not reflect all potential downstream use-cases - for example, we do not test multilingual performance. As such, examining how our results carry to multilingual settings or greatly differing base models is interesting future work. However, we have done our best to explore a wide variety of evaluations and datasets, using publicly available resources to aid in reproducibility. In terms of computational efficiency, we note that, while PPO does well, it comes with a significantly increased computational cost compared to DPO. We only consider performance in this study and don't explicitly measure the relative computational cost of different methods (e.g., by measuring FLOPs cost of training with DPO vs PPO). There are also other aspects to keep improving PPO (e.g., doing new rounds of preference annotations to update the reward model for the changing distributions of the policy model), which we cannot exhaust in this paper, and we leave the exploration of them for future work.

**Broader Impacts.** Language models have recently been deployed extensively, but there are few public studies on the impact of different algorithms for learning from preference feedback and datasets on these models. We hope to shed light on the impact of this stage of LM training and aid in improving future LMs. We explicitly consider safety evaluations as part of our evaluation and hope that our findings help inform how to improve the safety of future LMs, limiting potential harms and enhancing potential benefits. However, we note that focussing too much on learning from single preference datasets may result in LMs being aligned to only relatively small subsections of the global population.

# B  Dataset Details

We examine the following datasets in this work. We provide details on where we source the data and the license associated with each dataset.

- **SHP-2** [15]: We use the publically available HuggingFace train split, and randomly down-sample to 500,000 samples. The stackexchange portion of the dataset is licensed under the CC BY-SA license, and the reddit portion made available in accordance with the Reddit API terms of use. See the HuggingFace dataset card for more details.
- **StackExchange** [25]: We use the publically available HuggingFace train split, and randomly downsample to 500,000 samples. The dataset is licensed under the CC by-SA license.
- **PRM800k** [28]: We use the data from the second phase of collection. We consider prompts where the model generations achieved the correct answer once and failed to find the right answer once. We then randomly choose one correct and one incorrect generation as chosen and rejected respectively. The dataset is licensed under the MIT license.
- **Chatbot Arena Conversations (Chatbot Arena 2023)** [64]: We use the publically available HuggingFace train split at `https://huggingface.co/datasets/lmsys/chatbot_arena_conversations`. We remove all multi-turn samples as these diverge after the first turn (but users still rate the entire conversation), and filter out ties. The prompts are licensed under CC-BY-4.0 and the outputs under CC-BY-NC-4.0.
- **Chatbot Arena Preferences (Chatbot Arena 2024)** [7]: We use the publically available HuggingFace train split at `https://huggingface.co/datasets/lmsys/lmsys-arena-human-preference-55k`. We remove all multi-turn samples as these diverge after the first turn (but users still rate the entire conversation), and filter out ties. The dataset is licensed under the Apache 2.0 license.
- **AlpacaFarm Human and GPT-4 Preferences** [14]: We use the publicly available hugging-face 'preference' splits for each dataset. The dataset is licensed under CC-BY-NC-4.0.
- **HH-RLHF** [4]: We use the publically available HuggingFace train split. The dataset is licensed under the MIT license.
- **HelpSteer** [56]: We use the publically available HuggingFace train split. We average the fine-grained scores (except verbosity) for an overall score to decide chosen and rejected pairs. The dataset is licensed under the CC-BY-4.0 license.
- **Capybara 7k** [12]: We use the publically available HuggingFace train split released by Argilla available at `https://huggingface.co/datasets/argilla/`

| Dataset | # Samples |
|---|---|
| UltraFeedback (FG) | 60,908 |
| Stack Exchange | 60,908 |
| HH-RLHF | 60,908 |
| HelpSteer | 9,270 |
| PRM800k | 6,949 |
| Nectar | 60,908 |
| **Total** | 259,851 |

Table 6: Size of the subsplits of our RM mixture data.

`distilabel-capybara-dpo-7k-binarized`. The dataset is licensed under the Apache 2.0 license.
- **Orca Pairs** [34]: We use the publically available HuggingFace train split cleaned by Argilla[11], as we found it had better performance in initial experiments. The dataset is licensed under the Apache 2.0 license.
- **Nectar** [66]: We use the publically available HuggingFace train split. The dataset is licensed under the Apache 2.0 license.
- **UltraFeedback** [11]: We use the split released by Argilla[12], and consider two versions: one where the chosen and rejected samples are chosen using the overall score (Overall/OV) and one where they are chosen using an average of the fine-grained scores (Fine-grained/FG). The dataset is licensed under the MIT license.

We additionally create splits of HH-RLHF, StackExchange, and Nectar downsampled to 60,908 (matching the size of our UltraFeedback split) examples for size-equal comparisons of algorithms across different dataset types (i.e., Table 2). We also filter out any malformed examples we find[13].

We additionally construct our 'mix' used for expanding the reward model training by a number of examples from each dataset as listed in Table 6. Overall, our 'mix' prefence set contains roughly 260k samples, over four times larger than just using UltraFeedback.

## C   Model and Code details

We primarily build off TÜLU 2 13B and 70B [22], which themselves are based on Llama 2 [52]. TÜLU 2 models are licensed under the AI2 low-risk license and available at `https://huggingface.co/collections/allenai/tulu-v2-suite-6551b56e743e6349aab45101`. Llama 2 is licensed under a custom license and available at `https://huggingface.co/collections/meta-llama/llama-2-family-661da1f90a9d678b6f55773b`.

For code, we build off EasyLM [18], specifically the fork used to train TÜLU 2 [23], which is licensed under Apache 2.

## D   Evaluation Details

We use the following evaluations. We provide the category we map each evaluation to in brackets.

- **MMLU (factuality)** [20]: We use the official MMLU evaluation script and prompts available at `https://github.com/hendrycks/test`, with modifications to allow for batch processing. We evaluate using 0 few-shot examples, following the original setup of MMLU. We report average accuracy across test examples.
- **GSM8k (reasoning)** [9]: We evaluate models on the test set of GSM. Following Wei et al. [57], we evaluate with chain-of-thought. We use 8 few-shot in-context examples. Because all answers in GSM are numbers, we extract the last number in the model response as the final answer. We report average accuracy across test examples.

---

[11] `https://huggingface.co/datasets/argilla/distilabel-intel-orca-dpo-pairs`
[12] Manually cleaned and TruthfulQA prompts removed.
[13] For example, conversations with empty turns.

- **Big Bench Hard (BBH; reasoning)** [5, 47]: We follow the setup described in the original paper and evaluate with chain-of-thought. The officially provided prompts, which have 3 few-shot in-context examples are used. For the CoT setup, we extract the first word after the phrase 'So the answer is', or the entire response if there is no such substring present. We report average accuracy over sub-tasks (all of which use accuracy as the primary metric).
- **TruthfulQA (truthfulness)** [29]: Following Touvron et al. [52], we mainly use the generation setting. We follow the official script in their official implementation[14] to do greedy decoding and answer postprocessing. We also follow their instruction to train two GPT-based classifiers to judge the truthfulness and informativeness of the model response. We report the % Informative and Truthful.
- **AlpacaEval (instruction following)** [27, 13]: We use the package provided by Li et al. [27], following the default setup for both AlpacaEval 1 and length-controlled AlpacaEval 2 [13]. We allow the evaluated model to generate up to 8192 tokens, without specifying special stop sequences.
- **IFEval (instruction following)** [65]: IFEval benchmarks whether models can follow instructions that contain verifiable constraints, such as "write in more than 400 words". We use the official evaluation code released with the original paper[15], and report the "Loose Accuracy" metric at the prompt level (i.e., a response is counted as correct only if all the constraints in the prompt are detected to be satisfied after normalizing the response).
- **HumanEval+ (coding)** [6, 32]: We use the augmented form of the HumanEval dataset introduced by Liu et al. [32], which contains additional test cases. We additionally use the instructions provided by HumanEvalPack [36] when prompting instruction-tuned models. We report pass@10 and sample with a temperature of 0.8. We use v0.2.1 of the released Eval+ package.
- **MBPP+ (coding)** [2, 32]: For coding, we also evaluate using MBPP+, for which we prompt models to complete a program program given a natural language description and function signature. Similar to HumanEval+, We report pass@10, sampling with a temperature of 0.8. We use the v0.2.1 of the released Eval+ package.
- **ToxiGen (safety)** [19]: We follow the setup in Touvron et al. [52], but use the original set of prompts from Hartvigsen et al. [19], which are designed to elicit toxic generations for certain groups. We take only the prompts designed to produce toxic language ('hateful' prompts) and use 500 prompts per group to reduce evaluation costs. For base language models, we pass in the original ToxiGen prompts unchanged and greedily decode up to the first new line (or a maximum of 512 tokens). For instruction-tuned models, we place the prompt in the corresponding template, and ask the model to complete the prompt, until the model generates a stop token (or a maximum of 512 tokens). We pass the generated text into a roberta-large model trained to detect toxic content finetuned as part of Hartvigsen et al. [19] (available at `https://huggingface.co/tomh/toxigen_roberta`). We then report the percentage of generations deemed toxic by the classifier.
- **XSTest (safety)** [42]: XSTest evaluates whether a model can correctly refuse to follow malicious instructions, while still being able to correctly identify similar but safe instructions and follow those ones correctly. We use the same set of test prompts provided in their official repository [16], which includes a mixture of 200 unsafe prompts and 250 safe prompts. Similar to the original paper, we tried using both GPT4 and heuristic rules to detect whether the model responses are refusal. We found that GPT4 provides more reliable detection, as modern models may respond in various ways that exceed the coverage of those pre-defined rules. For GPT4-based detection, we found the originally provided prompt in the official repository has a confusing definition of "full refusal" and "partial refusal", so we removed that distinction and rewrote the prompt. In this paper, we report the F1 metric, which aggregates the precision and recall of model refusals. We will release all our evaluation codes for reproduction.

| | # Samples | Factuality | Reasoning | Coding | Truthfulness | Safety | Inst. Foll. | Average |
|---|---|---|---|---|---|---|---|---|
| Llama 2 base | - | 52.0 | 37.0 | 30.7 | 32.7 | 32.7 | - | - |
| Tulu 2 (SFT) | - | 55.4 | 47.8 | 45.1 | 56.6 | 91.8 | 44.2 | 56.8 |
| UltraFeedback (FG; All) | 60.908 | 55.3 | **50.9** | 45.9 | **69.3** | 91.9 | **52.8** | **61.0** |
| Strong Model Gen.s | 10,000 | 55.3 | 46.5 | 45.6 | 58.9 | 90.1 | 47.6 | 57.3 |
| Middle Model Gen.s | 10,000 | 55.2 | 48.5 | 44.5 | 60.6 | 91.5 | 48.2 | 58.1 |
| Weak Model Gen.s | 10,000 | 55.2 | 47.3 | 45.4 | 60.5 | 90.4 | 46.2 | 57.5 |
| FalseQA Prompts | 2,339 | 55.2 | 49.2 | 44.0 | 60.8 | **92.2** | 45.5 | 57.8 |
| Evol Instruct Prompts | 10,000 | **55.5** | 47.8 | **47.5** | 58.4 | 90.6 | 48.7 | 58.1 |
| TruthfulQA Prompts | 811 | 55.4 | 47.6 | 44.5 | 55.7 | 91.6 | 45.4 | 56.7 |
| Flan V2 Prompts | 2,0939 | 55.3 | 49.4 | 46.3 | 62.4 | 91.3 | 48.9 | 58.9 |
| ShareGPT Prompts | 19948 | 55.4 | 49.8 | 43.9 | 60.2 | 87.5 | 50.3 | 57.9 |
| UltraChat Prompts | 9929 | 55.4 | 50.6 | **47.5** | 59.9 | 89.9 | 48.1 | 58.6 |

Table 7: Performance of varied sub splits of UltraFeedback, considering both samples using generations from models of varying strength (weak, middle, strong model gen.) strong and only considering samples using prompts from single datasets (e.g., FalseQA, Flan V2). Sampling from different-quality models makes relatively little difference, while prompt choice matters. Additionally, different source datasets provide improvements in different evaluations.

# E  UltraFeedback Subset Study

Intrigued by the strong performance of UltraFeedback, the overall best performing dataset, we further compare the performance of differing subsets of UltraFeedback, and show our results in Table 7. First, we compare using *only* generations from the highest-scoring, middle-scoring, and lowest-scoring models (as judged by GPT-4). Surprisingly, we find that there is little difference in overall performance between these splits. This suggests the preference annotations from GPT-4 (i.e., the decisions about chosen and rejected pairs) are more important than the strength of the underlying models used to sample responses. Then, we compare models only trained on samples from the same original dataset (as UltraFeedback was constructed by sampling prompts from various sources). We find that each prompt source performs best in at least one category. This suggests that selecting a diverse set of prompts when generating synthetic data is important, and UltraFeedback already contains a diverse set of prompts covering various downstream applications.

**Subset Construction Details.** We construct the 'top', 'middle', and 'bottom' sets of UltraFeedback by using the average score of responses from each model according to the average of fine-grained scores. We then bucket the models into three groups accordingly. We provide the groups and average scores in Table 8. For constructing the splits, we then filter UltraFeedback to only include responses from the given models, pick the highest-scoring response per prompt as the chosen, and a random lower-scoring prompt as rejected. We construct a subset of 10,000 prompts for each group of models. For the prompt subsets, we simply use source dataset annotations provided in the UltraFeedback data itself.

# F  Additional Details for PPO and DPO

We performed hyperparameter search for both DPO and PPO, sweeping across core hyperparameters in pilot experiments on HH-RLHF and UltraFeedback. We provide further details for both algorithms below.

## F.1  DPO

We testing multiple values for $\beta$ (0.1, 0.01, 0.001) and for learning rate (5e-6, 5e-7, 5e-8) in pilot experiments on HH-RLHF and UltraFeedback. Ultimately, we found that the hyperparameters suggested by Tunstall et al. [53] and followed by Ivison et al. [22] worked best (learning rate of 5e-7, $\beta$ of 0.01). This additionally involves 3 epochs of training with a learning rate of $5 \times 10^{-7}$, with a

---

[14]https://github.com/sylinrl/TruthfulQA/

[15]https://github.com/google-research/google-research/tree/master/instruction_following_eval

[16]https://github.com/paul-rottger/exaggerated-safety

| Group | Model | Avg. Score |
|---|---|---|
| Top | gpt-4 | 4.50 |
| | gpt-3.5-turbo | 4.48 |
| | wizardlm-70b | 4.14 |
| | bard | 4.12 |
| Middle | vicuna-33b | 3.95 |
| | mpt-30b-chat | 3.92 |
| | llama-2-70b-chat | 3.91 |
| | wizardlm-13b | 3.91 |
| | llama-2-13b-chat | 3.78 |
| | ultralm-65b | 3.69 |
| | ultralm-13b | 3.63 |
| Bottom | wizardlm-7b | 3.44 |
| | llama-2-7b-chat | 3.39 |
| | starchat | 3.13 |
| | alpaca-7b | 2.97 |
| | pythia-12b | 2.60 |
| | falcon-40b-instruct | 2.57 |

Table 8: Model splits and average UltraFeedback fine-grained score used for construct UltraFeedback subsets used in Table 7.

linear warmup for 10% of training and linear cooldown for the duration. We use the open-sourced codebase used for training TÜLU 2+DPO.[17]

### F.2 PPO

**More on the algorithm.** PPO formulates language generation as a Markov decision process (MDP), where each response $y$ is an episode, an action is a token $y_t$, and a state is the concatenation of a prompt and a partial response ($x \circ y_{<t}$). To construct token-level rewards $r_t$ that guide the RL training, the sequence-level reward $r$ is applied only to the last token, and each token is penalized by a KL term $-\beta\big(\log \pi_\theta(y_t \mid x \circ y_{<t}) - \log \pi_{\text{ref}}(y_t \mid x \circ y_{<t})\big)$. Note that $\pi_\theta$ is the model we are training and $\pi_{\text{ref}}$ is a reference model (in our case, the initial SFT model we start training from). Formally, the token-level reward $r_t$ for each response token $y_t$ is defined as

$$r_t = \begin{cases} -\beta\big(\log \pi_\theta(y_t \mid x \circ y_{<t}) - \log \pi_{\text{ref}}(y_t \mid x \circ y_{<t})\big) & (\text{where } 1 \leq t < |y|) \\ -\beta\big(\log \pi_\theta(y_t \mid x \circ y_{<t}) - \log \pi_{\text{ref}}(y_t \mid x \circ y_{<t})\big) + r & (\text{where } t = |y|) \end{cases} \quad (4)$$

In addition to the policy model, PPO also trains a value model $V_\phi(x \circ y_{<t})$ that estimates the expected value function of incomplete responses under the active policy. The value model typically shares the same architecture as the reward model, while the regression head can be applied on any response token. The value model helps with estimating the *advantage* on each token, $A_t = -V_\phi(x \circ y_{<t}) + G_t$, where $G_t = \sum_{t'=t}^{|y|} \gamma^{t'-t} r_{t'}$ is the empirical return.[18] PPO trains the policy model by minimizing loss[19]

$$\mathcal{L}_\pi(\theta) = -\mathbb{E}_{x \in D_\pi, y \sim \pi_\theta(y|x), t \in [1,|y|]} \left[ \frac{\pi_\theta(y_t \mid x \circ y_{<t})}{\pi_{\text{ref}}(y_t \mid x \circ y_{<t})} \cdot A_t \right], \quad (5)$$

and trains the value model by minimizing the MSE loss against the empirical return:

$$\mathcal{L}_V(\phi) = \mathbb{E}_{x \in D_\pi, y \sim \pi_\theta(y|x), t \in [1,|y|]} \left[ \frac{1}{2} \big(V_\phi(x \circ y_{<t}) - G_t\big)^2 \right]. \quad (6)$$

---

[17] https://github.com/hamishivi/EasyLM

[18] In practice, a generalized advantage estimation is used.

[19] In practice, we clip the ratio $\nu_t(\theta) = \frac{\pi_\theta(y_t|x \circ y_{<t})}{\pi_{\text{ref}}(y_t|x \circ y_{<t})}$ by $1 \pm \epsilon$, and minimize $\mathcal{L}_\pi(\theta) = -\mathbb{E}\big[\min\big(\nu_t(\theta) \cdot A_t, \text{clip}(\nu_t(\theta), 1 - \epsilon, 1 + \epsilon) \cdot A_t\big)\big]$

The models can be jointly optimized by linearly combining the two losses: $\mathcal{L}_{\text{PPO}}(\theta, \phi) = \mathcal{L}_{\pi}(\theta) + \alpha \cdot \mathcal{L}_V(\phi)$.

**Implementation details.** PPO comes with many implementation details. We made a simplistic implementation that results in stable training. Notably:

- We initialize the value model from the reward model. This follows from InstructGPT [37]. Some other implementations initialize from the SFT model [33] or the base model [30, 58], while replacing the LM head with a regression head.
- For truncated completions, we set the reward to a large negative number (e.g., -10.0), which is referred to as the **EOS trick**.
- We do not perform normalization on the rewards. This follows from AlpacaFarm [14]. Although reward models trained under different settings can have very different output ranges, we found our PPO experiments quite robust to such variation.
- We do not whiten the step-level rewards within each batch. We do whiten the step-level advantages within each batch, following other implementations.
- We use a fixed KL penalty coefficient. The original PPO algorithm [44] has an adaptive KL controller, but most recent implementations have moved away from this [37, 52].

See Table 10 for a comparison with other open-source implementations.

**Hyperparameters.** The prompt and continuation each has at most 1024 tokens. We use a batch size of 64 and the same for minibatch size, and train for 1 epoch over the prompt dataset. On each batch of prompts and rollouts, we train for 1 inner epoch. We use a sampling temperature $\tau = 0.7$ for rollout, and a fixed KL penalty coefficient $\beta = 0.05$. We optimize with AdamW with learning rate $\eta = 1 \times 10^{-6}$ with a linear warmup for 10% of iterations. We provide additional details on the hyperparameters used for PPO in Table 9. In experiments, we found that training with larger (70B) reward models benefited from a reduced KL penalty coefficient $\beta$, but using smaller reward models did not benefit (see App I for further discussion). As such, for runs using larger reward models, we use a KL penalty of $\beta = 0.0325$ instead of $\beta = 0.05$. This is in line with prior work suggesting that improved reward models are more difficult to overoptimize against, and so can benefit from lower KL penalties [17, 40]. We found these hyperparameters through experimenting in pilot experiments over HH-RLHF and UltraFeedback. We experimented with KL penalty coefficients of 0.01, 0.025, 0.0325, 0.05, as well as varying the batch size (including taking more samples per gradient step, and taking multiple steps over the same generations). We additionally explored training for up to 3 epochs and found that training beyond 1 epoch did not yield improved performance and occasionally resulted in the training collapsing entirely, and as such stuck to training for only 1 epoch.

We additionally provide a more detailed comparison of our PPO implementation to other work in Table 10.

**Reward model hyperparameters.** For training the reward model, we follow prior work [11, 37] in only training the reward model for one epoch, using a learning rate of $1 \times 10^{-5}$ that warms up for the first 3% of training steps and linearly decays to $1 \times 10^{-6}$ by the end of training. We use a batch size of 512.

## G   Additional Observations with PPO

**Choice of hyperparameters.** We found that training beyond one epoch did not yield improved performance and occasionally resulted in the training collapsing entirely, and thus only trained for 1 epoch in our experiments ($E = 1$ in Table 9). We also found that training multiple inner epochs on each batch destabilizes PPO training, and thus we use 1 inner epoch ($e = 1$).

**Accelerating training with larger prompt batch size.** Rollout (i.e., generating online responses from the policy) is the most time-consuming step in PPO, taking up more than 95% of the total training time. We explored speeding up training by increasing the prompt batch size from 64 to 512, such that we can obtain more responses with a small latency overhead. We kept the minibatch size at 64 due to memory constraint, and because we need about 900 gradient steps to make the training converge.

This setup implies that rollouts in later minibatches become slightly off-policy when the model trains on them. In most implementations of PPO, the forward passes are also done on the batch level and

| Symbol | Value | Description |
|---|---|---|
| | | TRAINING |
| $B$ | 64 | Prompt batch size (for rollout). |
| $r$ | 1 | Number of rollouts for each prompt. |
| $b$ | 64 | Minibatch size for forward-backward passes. |
| $g$ | 1 | Gradient accumulation (in number of minibatches). |
| $E$ | 1 | Training epochs. |
| $e$ | 1 | Inner epochs trained for each batch. |
| $L_p$ | 1024 | Max number of tokens in the prompt. |
| $L_c$ | 1024 | Max number of tokens in the continuation. |
| | | RL |
| | 0.0 | Dropout rate. |
| $\tau$ | 0.7 | Temperature for sampling rollouts. |
| $\beta$ | 0.05, 0.0325 | KL penalty coefficient. |
| $\gamma$ | 1.0 | Discount factor for rewards. |
| $\lambda$ | 0.95 | Parameter for generalized advantage estimation. |
| $\varepsilon$ | 0.2 | Clipping range for policy and value losses. |
| $\alpha$ | 0.1 | Value loss coefficient. |
| | | OPTIMIZATION (ADAMW) |
| $\eta$ | $1 \times 10^{-6}$ | Learning rate. |
| | 10% | Percentage of iterations for warmup. |
| $(\beta_1, \beta_2)$ | (0.9, 0.95) | AdamW optimizer hyperparameters. |
| $\epsilon$ | $1 \times 10^{-5}$ | AdamW optimizer hyperparameter. |
| | 0.0 | Weight decay. |
| | 1.0 | Max global norm for gradient clipping. |

Table 9: PPO hyperparameters. We use values listed here unless otherwise noted.

| | Quark | Rainier/Crystal | FG-RLHF | AlpacaFarm | Ours |
|---|---|---|---|---|---|
| Init value from reward | ✗ | ✗ | ✗ | ✓ | ✓ |
| EOS trick | ✗ | ✗ | ✗ | ✗ | ✓ |
| Reward normalization | ✓ | ✓ | ✓ | ✗ | ✗ |
| Reward whitening | ✓ | ✓ | ✓ | ✓ | ✗ |
| Advantage whitening | ✓ | ✓ | ✓ | ✓ | ✓ |
| Adaptive KL controller | ✓ | ✗ | ✗ | ✗ | ✗ |
| KL clamping | ✗ | ✗ | ✗ | ✓ | ✗ |
| Multiple rollouts per prompt | ✗ | ✗ | ✓ | ✗ | ✗ |

Table 10: Variations in implementation of the PPO algorithm. Compared with serveral open-source PPO implementations: Quark [33], Rainier [30], Crystal [31], FG-RLHF [58], and AlpacaFarm [14].

thus the resulting logprobs and values are also obtained from a slightly stale version of the models. In our experiments, this severely destabilized training, and a closer investigation shows that the value model loss did not converge fast enough to supply policy training with reliable advantage signals. As a remedy, we made the forward passes to be carried out on the minibatch level so that these logprobs and values are obtained from the most current models. Note that this slightly deviates from most PPO implementations.

In Table 11, we compare the training time and performance of our default PPO setup and the larger prompt batch size setup. Increasing the prompt batch size by 8x reduces the training time by 5x (60 hours → 12 hours), while also decreasing the overall performance by 0.6% (61.7% → 61.1%). To remedy the performance loss, we experimented with generating multiple (up to 4) rollouts for each prompt, which increased the effective batch size for gradient updates while keeping the total number of gradient steps fixed. When using a rollout multiplier of 4, most performance loss can be recovered, while the training speedup is also less dramatic.

Our conclusion from this set of experiments is that, it is important for the forward passes to be performed online, and to get the extra mile offered by PPO, the rollouts should also be generated fully online. Deviating from this may speed up training, but at some performance cost. Since we

| $B$ | $b$ | $r$ | $g$ | # training ex. | grad update bsz | # grad updates | **Training Time** | **Avg. Perf.** |
|---|---|---|---|---|---|---|---|---|
| 64 | 64 | 1 | 1 | 1x | 1x | 1x | 60 h | 61.7 |
| 512 | 64 | 1 | 1 | 1x | 1x | 1x | 12 h | 61.1 |
| 512 | 64 | 2 | 2 | 2x | 2x | 1x | 16 h | 61.1 |
| 256 | 64 | 4 | 4 | 4x | 4x | 1x | 32 h | 61.4 |

Table 11: Performance of PPO under bigger prompt batch size. The first row uses the same experiment setup as the PPO model trained on UltraFeedback (FG), as in Table 2. Hyperparameter notations are same as Table 9: $B$ = prompt batch size, $b$ = minibatch size for forward-backward passes, $r$ = # rollouts per prompt, $g$ = gradient accumulation. Number of training examples, gradient update batch size, and total number of gradient updates are relative to the first row. We keep the total number of gradient updates fixed, train all models for 1 epoch. Increasing the prompt batch size can speed up PPO training at some performance cost, and most performance loss can be recovered by increasing $r$ and $g$ (which effectively increases the gradient update batch size).

| | GSM | BBH | Codex-Eval+ | MBPP+ | AEval 1 | AEval 2 | IFEval | XSTest |
|---|---|---|---|---|---|---|---|---|
| **Model** | | | | | | | | |
| Tulu 2 13B (SFT) | 46.0 | 49.5 | 19.3 | 38.1 | 78.9 | 5.0 | 43.4 | 85.3 |
| 13B UltraF. RM | 60.5 | 50.0 | 29.3 | 39.9 | 91.6 | 20.5 | 47.1 | 84.6 |
| 13B Mix RM | 60.5 | 54.3 | 29.3 | 39.9 | 92.7 | 22.3 | 50.3 | 86.2 |
| 70B UltraF. RM | 67.0 | **59.9** | **37.2** | **41.4** | 92.6 | 22.4 | **51.0** | **86.3** |
| 70B Mix RM | **67.5** | 59.8 | 35.4 | 38.5 | 90.9 | 19.0 | 46.6 | 85.6 |

Table 12: Full results from Best-of-N evaluation summarized in Table 3.

didn't get better results with the large prompt batch size setting, we did all other experiments with the default prompt batch size of 64.

# H Reward Model Evaluation Details

**Best-of-N Details** For best-of-N, we sample 16 responses from TÜLU 2 13B with a temperature of 0.7 for each evaluation task we examine. We then pass these responses (along with the prompt used for generation) into the given reward model, and use the top-scoring response as the final output.

Given a list of these top-scoring outputs, we then pass these to the rest of the given evaluation setup. We examine only a subset of evaluations, focussing on the evaluations that rely on long-form model generations (as we are most interested in the reward model's ability to judge these outputs during PPO training). In particular, we look at GSM8k, BBH, Codex-Eval+, MBPP+, AlpacaEval 1 and 2, IFEval, XStest. For Codex-Eval+ and MBPP+, we use Pass@1 instead and just evaluate the top-scoring example (unlike other tables in which we use Pass@10). When reporting the average score in Table 3, we first calculate an average score per evaluation task category (following the same evaluation categories as in App. D), and then report the average over categories.

We report the full best-of-N results across each evaluation in Table 12.

| | Chat | Chat Hard | Safety | Reasoning | Prior Sets | Score |
|---|---|---|---|---|---|---|
| 13B UltraFeedback RM | 74.3 | 49.3 | 52.2 | 65.0 | 67.9 | 61.0 |
| 13B Mix RM | **97.2** | **61.2** | **85.9** | **78.1** | **73.7** | **79.8** |
| 70B UltraFeedback RM | 96.4 | 60.5 | 63.7 | 74.8 | 71.4 | 73.6 |
| 70B Mix RM | 94.4 | 52.2 | 83.3 | 65.9 | 73.5 | 73.9 |

Table 13: Full results from RewardBench evaluation for results shown in Table 3. Score is a weighted average of subsets with prior sets given weight 0.5 and all other sets given weight 1, following Lambert et al. [26].

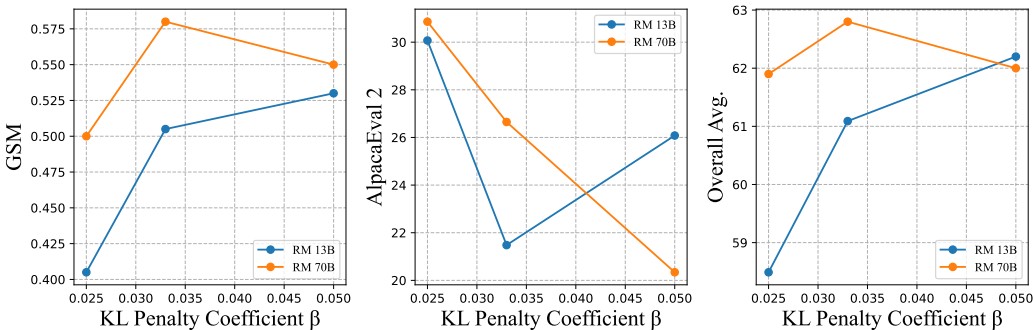

Figure 4: Performance of models trained with 13B and 70B UltraF. RMs with varying KL coefficient values. (a) GSM Accuracy, (b) AlpacaEval 2 winrate, (c) Overall performance across entire evaluation suite. Overall best performance occurs at different KL coefficient values for different reward models. However, AlpacaEval 2 performance grows with reduced KL coefficient values.

**RewardBench Details** We report the subset scores of our reward models on RewardBench in Table 13.

## I   KL Penalty Coefficient Exploration

As seen in Figure 4, we observe that the optimal KL penalty coefficient ($\beta$ in Eq. 2) value changes with the reward model used. While using the smaller 13B reward model results in large drops in overall performance as $\beta$ shrinks, the larger 70B RM is more robust and actually achieves higher performance at a smaller $\beta$ than the 13B UltraF. RM. This is in line with prior work suggesting that larger reward models are less prone to overoptimization [17, 10], and as such are more robust to lower $\beta$ (which encourages more optimization against the RM). This highlights an additional potential benefit of larger RMs: they may be easier to tune due to their increased robustness to choices of $\beta$.

We also note that some evaluations do not suffer as much from reduced $\beta$. For example, AlpacaEval 2 performance (Fig. 4c) actually is highest at the lowest $\beta$ shown. This is likely due to the match between AlpacaEval and PPO training: as the reward models are trained on GPT-4 preferences, more closely optimizing against them (via a lower $\beta$) also improves the rate at which the model generations are preferred *by GPT-4*. However, it also comes with significant reductions in other evaluations (as seen in the drop in GSM performance in Fig 4), which is undesirable when training a generalist multi-task model. This is in line with prior work observing that there appears to be an 'alignment tax' (i.e., reduced scores on various capabilities including reasoning and math) when performing learning from preference feedback [37, 4], and that this tax can be partially controlled through tuning $\beta$ (e.g., Fig. 34 in Ouyang et al. [37]).

## J   Prompt Domain Identification Details

In order to find additional unlabelled prompts, we mine prompts from UltraFeedback [11], Wild-Chat [62], and LMSYS-1M [63]. We only consider the prompts and throw away the accompanying model responses. We categorise unlabelled prompts by prompting TÜLU 2 70B. We first prompt TÜLU 2 70B to tag the prompts with various categories using the prompt shown in Figure 5. We then sample Code and Math prompts by picking prompts that include 'Coding' or 'Math' tags and as few other tags as possible. We remove all prompts with 'unclear' and 'multilingual' tags as we wish to focus on clear queries made in English. The authors examined a small portion of the mined prompts by hand and found this resulted in a solid collection of high-quality prompts for math and code respectively.

```
# Instruction

Please label the task tags for the user query.

## User Query
‘ ‘ ‘
{$instruction}
‘ ‘ ‘

## Tagging the user input

### Task Tags

all_task_tags = [
    "Coding",  # Users seek help with writing, reviewing, or fixing code in programming.
    "Math",  # Queries related to mathematical concepts, problems, and calculations.
    "Asking for Advice",  # Users seek recommendations, suggestions,
    or advice on various topics.
    "Brainstorming",  # Involves generating ideas, creative thinking,
    or exploring possibilities.
    "Classification",  # Queries require categorizing or organizing information into
    groups or classes.
    "Closed Question Answering",  # Users ask questions that require a specific
    answer or a short response.
    "Creative Writing",  # Users seek assistance with crafting stories, poems,
    or other creative texts.
    "Extraction",  # Involves extracting specific information or details from
    a larger body of text.
    "Inhabiting a Character/Persona",  # Users engage in scenarios requiring
    the model to adopt a character or persona.
    "Open Question Answering",  # Users ask questions that require detailed
    or elaborate responses.
    "Rewriting",  # Users ask for help in rephrasing, summarizing, or rewriting text.
    "Summarization",  # Involves condensing information, text, or content
    into a shorter form.
    "Multilingual",  # Queries involving non-English natural languages.
    "Unclear",  # Queries that are ambiguous, vague, or unclear.
]

## Output Format

Note that you can only select the most relevant task types.
Please use the multilingual tag if the query is in a language other than English.
Add the unclear tag if there is no obvious question to answer in the prompt.
Now, please output your tags below in a json format by filling in the placeholders in []:
‘ ‘ ‘
{
    "tags": ["[tag 1]", "[tag 2]", ... ]
}
‘ ‘ ‘
Do not add any additional characters.
```

Figure 5: Prompt used for classifying unlabelled prompts.

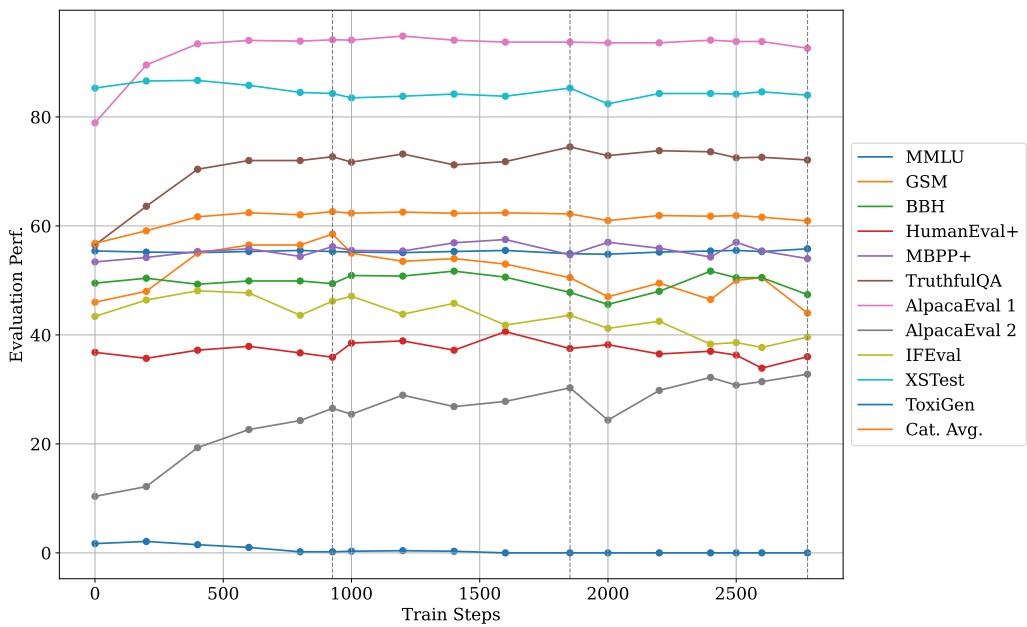

Figure 6: Performance of all evaluations over PPO training steps when training using the 70B UltraFeedback RM and UltraFeedback prompts for 3 epochs. Grey dashed lines indicate epoch boundaries.

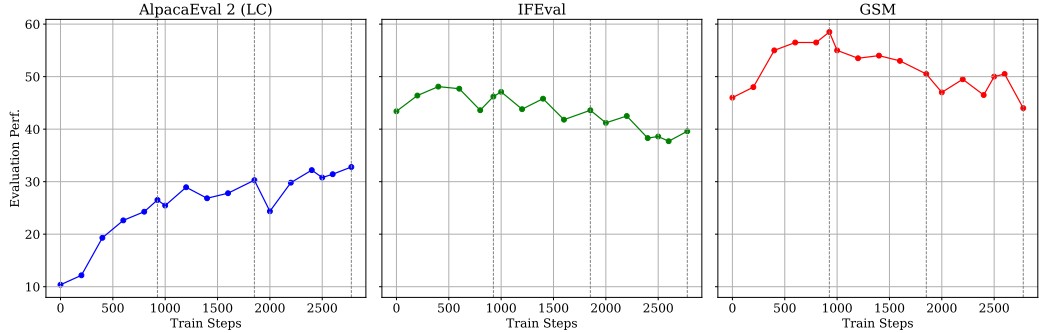

Figure 7: Performance on (left) AlpacaEval 2, (middle) IFEval, (right) GSM8k over PPO training steps when training using the 70B UltraFeedback RM and UltraFeedback prompts for 3 epochs. Grey dashed lines indicate epoch boundaries.

## K    Compute Details

We train all models on 256 or 128-size TPUv3 pods. PPO training with a 13B reward model and 13B policy on the UltraFeedback set takes 3 days on a 256-size pod, with the reward model training taking roughly 10 hours to train. DPO training on UltraFeedback (roughly 60,000 samples) for a 13B model takes 9 hours. We note that we conducted additional smaller-size runs that we do not report results from in this paper as part of our research to further explore dataset and hyperparameter choices.

## L    Model Performance over Training

We investigate how the performance of our best model (PPO training TÜLU 2 13B with the 70B UltraFeedback RM and UltraFeedback prompts), and continue training beyond the 1 epoch result reported in the main text of the paper. We show these evaluations in Figure 6. We find that how performance changes over training is quite distinct between each evaluation: while some evaluations

continuously improve (e.g., AlpacaEval 2), others improve and then degrade (IFEval, GSM8k), or remain relatively unchanged over training (MMLU). We show the AlpacaEval 2, IFEval, and GSM8k performances individually in Figure 7. This highlights the need to measure a *broad variety of benchmarks*: while just examining AlpacaEval 2 would suggest that training for 3 epochs (or longer) is best, examining IFEval we would find that the model after 500 steps (0.5 epochs) is best, and GSM8k peaks at the 1 epoch mark. As such, we also broadly caution against evaluating model performance only on llm-as-judge benchmarks such as AlpacaEval, as we observe that improved AlpacaEval performance may come at the cost of other desirable properties.

