# OpenReview forum: "Unpacking DPO and PPO: Disentangling Best Practices for Learning from Preference Feedback"
_NeurIPS.cc/2024/Conference — NeurIPS 2024 poster_

### Official Review · Reviewer_P8L3 · 2024-06-13

**Soundness:** 2
**Presentation:** 4
**Contribution:** 2
**Rating:** 4
**Confidence:** 4

**Summary:**

The paper identifies four core components in training from preference feedback (RLHF): labeled preference data, learning algorithm, reward model, and unlabeled training prompts; and conducts a study on the effect of each component separately to disentangle their contributions to performance.
The authors experiment with variations of each component and make several claims:
1. There is a clear ordering of the impact of each component on performance: preference data quality > algorithm choice > reward model quality > targeted policy prompts.
2. Synthetic data with (aggregated) per-aspect preferences works best.
3. PPO outperforms DPO
4. Increasing the reward model size or its training dataset size improves its performance on reward benchmarks, but has marginal effects on downstream tasks overall.
5. targeted online training prompts can improve performance in domain-specific settings, but not overall performance.

Hence, they suggest a recipe with the best of each component.

**Strengths:**

The paper is well presented: the claims are clear, the experiments backing them are easy to identify, and the narrative is easy to follow.

The description of the experimental setup is comprehensive and very detailed, helping the applicability of the paper.

The authors experiment with a comprehensive collection of preference datasets and benchmarks.

The authors make several interesting observations from their experiments such as the effects f the preference datasets used showing mostly on dimensions such as truthfulness and instruction following, or the improvement of reward modeling showing in rewards benchmarks but not translating to downstream tasks.

**Weaknesses:**

I find that the claims made in the paper often deviate from the experimental support provided, they are either overstated compared to the scope or the experiments, or inaccurately reflect the scope of the experiments. Specifically,

**Algorithm choice:**

Although comparing components such as prompt datasets is possible when keeping all else the same (algo, hyperparameters, etc), the claim "PPO outperforms DPO" cannot be made without carefully considering hyperparameters.
The authors have a single hyperparameter configuration for each model, and although they may have been validated in other settings, they have not been optimized (within a fixed but same budget) for each algorithm on the author's setting.
The DPO hyperparameters were originally used by Zephyr-Beta, and taken without tuning for TULU 2 [Appendix F.1, 22].
For PPO, apart from the number of epochs that have been cut to 1, it's not clear how the authors obtained the rest of the hyperparameters.

Furthermore, from Table 2, "PPO outperforms DPO" _on average_ on _pre-selected_ datasets. When looking at individual benchmarks DPO models outperform PPO models in quite a number of tasks, and Table 2 only includes a subset of datasets which has not been motivated in the paper.

Therefore I believe that the claim is misleading.

**Preference dataset**:

Only Table 1 with DPO is used to formulate the conclusions on the best preference dataset. For the claim to hold in the scope of the paper (considering its limitations with a single pre-trained model) the preference datasets have to be compared with at least multiple algorithms (and ideally hyperparameters).
Table 2 provides some results for PPO but on all datasets, so it fails to complement Table 1.

The authors do not provide insights about why synthetic data performs best. It is appreciated to observe such difference in performance, but the claim would have more value and impact with some analysis or intuition as to why it would hold.

The claim "Preference-based learning has the strongest effect on instruction following and truthfulness performance." may only be valid in when the preference data itself mainly includes preferences based on instruction following and truthfulness, which is the case in the datasets considered in the paper. However, it may be very valid, but inefficient, to train factuality using preferences.
The scope of such a claim should be carefully explicited.

**Reward model**:

Although the paper aims to disentangle the effects of each component in RLHF, the experiments it conducts on some components tend to depend on the results from the previous components.
E.g. the reward model experiments are only carried out on the preference datasets providing the best performance, giving a biased view on the effects of the reward model and does not allow to formulate an ordering on the performance contribution of each component independently.

Also, the average improvement of 0.6 from the 13B to the 70B UltraF. reward model has been qualified as marginal while the improvement of 0.7 from DPO to PPO has been qualified as significant.

**Questions:**

How did the authors select the preference datasets to compare DPO vs PPO in Table 2? One would expect to have them compared on all datasets as in Table 1 or to have a justification for the datasets that have been dropped/kept.

**Limitations:**

Although in the appendix, the authors adequately state the limitations of their work and its broader impacts.

---

> ### Author Rebuttal · Authors · 2024-08-07
>
> Thank you for your review, and noting that our work is clearly written, with an easy to follow narrative, detailed experimental setup, and multiple interesting observations. We clarify key points of our experimental setup and choices (e.g. datasets) below, which hopefully address your key concerns. We will add these additional details and clarifications to our updated paper.
>
> **Concerns**:
>
> 1. **Algorithm choice & Hyperparameter choice**
>
> In particular, for DPO, we tested values of {0.1, 0.01, 0.001} for beta {5e-6, 5e-7, 5e-8} for LR for both HH-RLHF and UltraFeedback. For PPO, we experimented with KL penalty coefficients of {0.01, 0.025, 0.0325, 0.05}, as well as varying the batch size (including taking more samples per gradient step, and taking multiple steps over the same generations). We discuss some of the details around our hyperparameter choices in Appendices F and G, and will explicitly add the details given above. For PPO hyperparameters we did not search extensively over (clip threshold, discounting factors, learning rates, adam betas), we borrow primarily from InstructGPT [2] and Llama-2 [3], and open-source repositories (Rainier [4], Finegrained-RLHF [5], AlpacaFarm [6]), which are common reference points for RLHF work. We provide additional notes on hyperparameter choice and additional observations using PPO in Appendices F and G. As such, we believe we made a reasonable attempt to tune both DPO and PPO hyperparameters independently, although we note that further performance gains could be made by tuning on a per-dataset basis.
>
> 2. **Dataset choice in Table 2**
>
> We apologize for not explaining the dataset choice in Table 2: we explicitly chose the top-performing dataset of each source from Table 1 (StackExchange from Web, HH-RLHF from human, Ultrafeedback from synthetic). We also include the second-best performing dataset overall (Nectar) and an additional human-annotated dataset from the popular evaluation platform (Chatbot Arena; we chose 2023 over 2024 to reduce computational cost). We will add these details to our updated manuscript.
>
> 3. **Why does synthetic data perform well?**
>
> We hypothesize that the synthetic data we test performs well for a few reasons: (1) annotation quality is generally high due to the use of GPT-4 as a judge [1] and GPT4 labels may be hypothetically more consistent, (2) the prompts chosen cover a wide range of topics useful for downstream evaluations (math, reasoning, chat, coding, etc.), (3) there is some alignment between synthetic preferences and our evaluation setting - for example, Chatbot Arena data is human-sourced, but shows a clear degradation in safety which hurts its overall performance, and using AlpacaEval in our evaluation means we have a slight bias to GPT-generated data (although this is not the case for the many other evaluations we examine). We believe that further exploring how well synthetically generated data compares to expert-level human annotations, and exploring differences between the two, is an interesting avenue for future work.
>
> 4. **Preference-based learning’s effect on factuality.**
>
> We agree and will adjust the wording to make it clearer that our observations are based on the datasets we consider in our introduction, conclusion, and section 3. We agree that a carefully-made factuality dataset may indeed improve performance (even if it is non-optimal compared to SFT-based approaches).
>
> 5. **Identifying the performance contribution independent of other components.**
>
> We agree that there may be ways in which the contribution of a different component varies based on the setting (for example, some datasets not tested may scale better than those tested for the reward model experiments). Due to computational constraints, we did not test every single possible combination of {rm size, prompt, dataset, hyperparameter choice}, but focussed on running ablations using the best choices from the previous steps (e.g., focussing on UltraFeedback for PPO ablations, and using the best datasets found for DPO for comparing to PPO).
>
> 6. **Marginal vs non-marginal improvement**
>
> We agree that both changes give improvements of similar size, although we believe that the PPO claim is slightly stronger due to being a result of tests across multiple datasets (in fact, the p-value for the observed difference is .04 under a two-tailed paired t-test), while the larger RM size increase in performance was only from one experiment. As such, we have less confidence in the claim that the larger RM helps, although the boost on GSM8k appears large. We will adjust the wording in each section accordingly to better match this (explicitly noting the statistical significance of the DPO & PPO result, and noting that the jump in performance from using the larger RM is of similar scale, but is from a single run and driven largely by GSM8k improvements).
>
> Additionally, we were somewhat surprised that a 10x increase in RM size (and almost 2x increase in best-of-N improvements, from 5.8 to 10.3) did not yield similarly large changes in the overall performance, and this coloured our reporting. We will adjust our wording accordingly (small changes in light of large BoN & compute increases).
>
> **Questions**:
>
> 1. **How did the authors select the preference datasets to compare DPO vs PPO in Table 2?**
>
> Please see our response to concern 2 above.
>
> ---
> [1] Zheng et al (2023). Judging LLM-as-a-judge with MT-Bench and Chatbot Arena. NeurIPS.
>
> [2] Ouyang et al. (2022). Training language models to follow instructions with human feedback. NeurIPS.
>
> [3] Touvron et al. (2023). Llama 2: Open Foundation and Fine-Tuned Chat Models. ArXiv.
>
> [4] Liu et al. (2022). Rainier: Reinforced Knowledge Introspector for Commonsense Question Answering. EMNLP.
>
> [5] Wu et al. (2023). Fine-Grained Human Feedback Gives Better Rewards for Language Model Training. NeurIPS.
>
> [6] Dubois et al (2023). AlpacaFarm: A Simulation Framework for Methods that Learn from Human Feedback. NeurIPS.

---

> > ### Comment · Reviewer_P8L3 · 2024-08-11
> >
> > I thank the authors for the clarifications. Their reply addresses all my points but also acknowledges a major limitation due to computational costs.
> >
> > I understand that the authors aimed to provide the best empirical recommendations they could given a limited budget and acknowledge that many of the numbers in the paper will be relevant to readers. However, I believe that under this limited computational budget, not enough research questions have been tackled and answered in satisfactory manner. I find this a necessary condition for a publication at the venue.
> >
> > I, therefore, increase my score but maintain it below the acceptance threshold.

---

> > > ### Author Response · Authors · 2024-08-13
> > >
> > > Thank you very much for your response and raising your score!
> > >
> > > As for research questions, we believe that we have examined a number of useful and interesting areas, including (a) what public datasets works well for DPO, (b) relative performance of DPO and PPO across different datasets (and models, with our Llama 3 results), (c) the effect (or rather, surprising lack thereof) of the size of the RM during PPO training (with similar results from increasing dataset size using well-performing datasets from prior steps), (d) the effect of using more domain-specific prompts during PPO training. We hope and believe these are interesting and useful results for researchers in the RLHF space, and supported by the experimental results we have reported.
> > >
> > > While it would be great to further explore these effects by doing a more thorough grid search over {dataset, RM size, prompt} combinations, we note that running a PPO experiment takes \~54 hours on a v3-256, and so running all possible dataset and RM size combinations (14 x 2) would take 1,512 hours (ignoring extra experiments exploring mixing datasets or varying prompt sets, which would incur further costs). Based on the google cloud calculator (https://cloud.google.com/products/calculator?hl=en) as of the 12th of august, 700 hours on a v3-256 (rather than newer, more expensive options) costs \\$394,240 in europe-west4, and so running these additional experiments would take \~\$800,000, and take 63 days to run (and if we wanted to e.g. explore the effect of using a different prompt set in each case, this would further multiply the cost). As such, we first ablated datasets in cheaper experiments (DPO), and then examined the more promising ones in PPO, and then additionally further explored key aspects of PPO using the best dataset found. This allows us to explore interesting and promising aspects of PPO without extreme computational costs.

---

> > > > ### Comment · Reviewer_P8L3 · 2024-08-13
> > > >
> > > > I totally understand the computational budget limits and did not imply that the budget was objectively low, nor did I imply that its use was not efficient. What I mean is that there is a misfit between the budget and the initial research question addressed or the claims expected from them, especially the "Disentangling" part, which didn't really end up being a disentangling.
> > > >
> > > > (a), (b), (c), and (d) can all be answered by observations and that's why I said that the numbers will be useful to readers, but I believe that they lack some research depth providing an understanding of why a number is better than another. The authors blamed the computational budget which I then raised as a limitation and a misfit, but I also believe that the current results can be analyzed further to answer other research questions with enough depth.
> > > > For this reason, I prefer to keep my score.

---

### Official Review · Reviewer_wF89 · 2024-07-05

**Soundness:** 3
**Presentation:** 4
**Contribution:** 2
**Rating:** 5
**Confidence:** 4

**Summary:**

The work summarizes the area of learning from preferences for optimizing language models. Specifically, the analyze four aspects: preference data, learning algorithm, reward  model, and policy training prompts. They empirically  answer questions on the downstream  improvement by improvements in each of these axis. Overall, they observe that the largest improvement arises from higher quality preference data, followed by choice of learning algorithm (PPO over DPO) with relatively smaller gains on reward model capabilities followed by training prompts.

**Strengths:**

* Clarity of writing: The work is well written, concise and clear to follow.
* Significance:  We gain practical insights from this work, including the use of synthetic datasets and large reward models, which can be valuable for practitioners in the field.
* Quality: Empirical results are well documented, methods are sound
* Originality: While this offers new insights, the work extends existing methodologies  rather than introducing novel concepts.

**Weaknesses:**

* Novelty: While the work helps answers some excellent questions, the overall scope of the work seems limited. It would be interesting if this could be expanded to a more comprehensive study, see the question below.

**Questions:**

How do these insights scale to other paradigms of language models such as in the  multimodal regime?

**Limitations:**

Yes, Section A in the Appendix.

---

> ### Author Rebuttal · Authors · 2024-08-07
>
> Thank you for your review and noting that our work is well written and clear to follow, and that the insights in our work are useful for practitioners in the field. We address concerns and questions below:
>
> **Concerns**:
>
> 1. **Novelty**
>
> While we do not propose entirely new methods ourselves, the novelty of this work is more on the comprehensive setup for comparing current resources for RLHF with LMs, and the findings supported by a large number of experimental results. To our knowledge, there is little prior work comparing and examining RLHF methods in detail with extensive empirical backing. Additionally, various findings in our work (e.g., the hardship in transferring reward model performance to policy model, the sensitivity to the prompt distribution in PPO) are the first instance of these observations/results being discussed publicly to the best of our knowledge. We believe such analysis should have novelty in itself and are valuable given the current status of LLM research, and believe that multiple reviewers do point out that our insights have practical use for researchers in the field.
>
> **Questions**:
>
> 1. **Extending to multimodal models?**
>
> In our work, we decided to focus on understanding popular methods and datasets across a range of evaluations, working on popular text-only models. Multimodal RLHF approaches are still very new compared to the language model space, and there are not as many available instruction tuning or preference datasets to explore. Nonetheless, we believe that exploring how our findings extend to the multimodal regime would be an interesting direction for future work.

---

> > ### Comment · Reviewer_wF89 · 2024-08-11
> > **Acknowledging Rebuttal**
> >
> > Thanks for taking the time answering the questions, keeping my score.

---

### Official Review · Reviewer_WxZd · 2024-07-07

**Soundness:** 3
**Presentation:** 3
**Contribution:** 3
**Rating:** 5
**Confidence:** 4

**Summary:**

This work concentrates on methods for LLM learning from preference feedback and conducts a lot of experiments to identify key aspects of the preference-based methods. The work gives an ordering for the  importance of the core aspects: preference data quality, algorithm choice, reward model quality, and finally targeted policy training prompt.

**Strengths:**

Since the pipeline of preference-based methods has many details, it is hard to see works like this one to have done plenty of ablation experiments. This work brings a lot of observations for this community and can help researchers to have more understanding on different parts of preference-based methods.

  This work gives an order for the importance of four key elements of preference learning. This is a relatively systematical investigation for reference-based learning.

  This work also gives a recipe for learning from preferences and this may guide better performance for LLM.

**Weaknesses:**

Although a lot of observations are provided in this work, it seems that authors don't provide a deeper and more systematic understanding of the whole process of preference learning. Indeed, this is not an easy task, but I think it would be really helpful if we could draw some deeper insights from all these observations.

  Since there are many  elements involved in the preference training process, it is not easy to control variates. I found that some results might be improved with more ablations.

**Questions:**

Here I have some detailed questions and I would be thankful if authors can give some further explanations.
1. On page 5 line 153, the quality of pairwise judgements is mentioned but there is not much explanation for this. Can we give a relatively quantitative method to judge the quality of pairs?
2. On page 5 line 161, it is mentioned that PPO outperforms DPO. Since there is a lot of DPO related research recently, I'm willing to see some more discussion about PPO and DPO. Is the offline property of DPO the main reason?
3. On page 6 table 3, it seems quite strange that 70B Mix RM has a lower RewardBench Score than 13B Mix RM. Is there any reason for this result?
4. On page 7 line 201, it is mentioned that there are no improvements in overall downstream performance. I noticed that the PPO is trained on UltraF prompts. Since the two Mix RMs are trained on more data, I might expect the two models can achieve better performance if the PPO is trained on the Mix prompts. In Fig.3, we observed that Mix RMs gain higher acc on GSM train. Also, I noticed that in table 4, Mix RMs trained on the code-and-math-mixed prompts have poor performance. Can we get some principles for dataset selection for RM?
5. On page 7 table 4, I found that the set of UF prompts and the set of Mixed prompts have the size. Since more code and math prompts are used in the Mixed prompts, some UF prompts are not used. Could this lead to the degradation of the Avg. Across all Evals? What if we use all UF prompts and some further code and math prompts to train a PPO?
6. On page 18 table 7, the UF dataset is split into Strong, Middle and Weak sets. Are the four datasets (UF all, Strong, Middle and Weak) downsampled to the same size for comparision?
7. On page 19 equation (5), is there any typo with the \pi_{ref}？Does it refer to the SFT policy?
8. The work shows that preference data is most important. I noticed that effect of dataset size is not well discussed. Is there any observation about this? Can more data provide more improvements?
9. I am curious about the training process during these experiments. For example, how the training reward changes, how the KL divergence between PPO policy and SFT policy changes, how the entropy of the policy changes, and how the length of the responses changes during the whole training processes. I would be grateful if authors could share some interesting observations during training.

---

> ### Author Rebuttal · Authors · 2024-08-07
>
> Thank you for your detailed review and questions. We address your concerns and questions below and hope this provides further insights into our work.
>
> **Concerns**:
> 1. **Results could be improved with more ablations.**
>
> Please see point 1 of our general response, where we additionally provide new results with a new base model (llama 3).
>
> 2. **It would be helpful to draw deeper insights.**
>
> Our aim in this work is primarily to explore and test the performance of popular RLHF methods on existing datasets, and is primarily empirical. We hope that our work serves as a useful starting point for further, more focused studies into RLHF (for example, investigating further why using much larger RMs does not lead to much larger gains). We also note that most reviewers agree our work does contain observations and results that are of interest to the broader community.
>
> **Questions**:
>
> 1. **The quality of pairwise judgements.**
>
> Quality here is primarily referring to the use of overall vs fine-grained judgements when deciding the chosen/rejected pairs in UltraFeedback, which has been shown to lead to improved reward model performance [1, table 2], even when using the same chosen/rejected pairs (and we find similarly improves DPO performance). We focus on the effect of data on downstream performance here, noting that it appears improving the chosen/rejected judgements can improve downstream performance.
>
> 2. **Why is PPO better?**
>
> We believe the offline vs online aspects of DPO and PPO are quite important for performance (as noted at the end of section 2.1), and there is concurrent work also suggesting this [2,3]. Additionally, we hypothesize that the RM being finetuned on preference data without any regularization terms (e.g. the beta in DPO) may allow it to more closely fit human preferences, which may result in a stronger policy downstream.
>
> 3. **Why does 70B mix rm have a lower RewardBench score than the 13B model?**
>
> RewardBench consists of outputs from varied models across varied prompts, and so may be somewhat out-of-domain compared to best-of-N (BoN) evaluations for the purpose of identifying good RMs for PPO. In contrast, our BoN evaluation indicates that both 70B RMs outperform the 13B models, suggesting that larger models are better at identifying superior outputs from the same initial policy used in PPO, which more directly relates to how the RM is used during PPO training. Examining BoN and RewardBench scores in more detail in Tables 12 and 13, we see that 70B mix RM outperforms the 13B mix RM on math (GSM), while degrading in instruction following (AEval 1 & 2, IFEval), which matches with the worse performance on the chat subsets of RewardBench. We find it surprising the 70B mix RM underperforms in the reasoning subset, considering the improved GSM performance, potentially due to differences in evaluation distributions (RewardBench reasoning subset consists mostly of coding questions).
>
> 4. **Can we get some principles for dataset selection for RM?**
>
> Figure 3’s best results are found when using the GSM8k train set as prompts, which is somewhat different to the typical zero-shot setting used for modern LMs, where we assume our models will be tested on tasks not explicitly selected for during training. We did try training our models with Mix RMs and mixture prompts for longer but still did not observe improvements in performance (see answer to Q5). Overall, the fact that RewardBench and BoN does not translate neatly to downstream performance from PPO makes determining good methods for dataset selection for RMs difficult. It would be interesting to further explore how to improve RM dataset selection with an eye towards downstream performance in future work.
>
> 5. **What if we use all UF prompts and some further code and math prompts to train?**
>
> We find that further training the model to use all the UltraFeedback prompts on top of additional math and code prompts does not result in consistent improved performance at the 13B scale, suggesting that using the full UF prompt with the mix prompts set does not provide consistent additional gains:
>
> | RM Type | avg perf with 60k prompts | avg perf with *all* UF + Mix prompts |
> |-|-|-|
> | 13B UltraF. RM |  61.9 | 61.3 |
> | 13B Mix RM | 60.9 | 61.4 |
>
> 6. **Are the four datasets (UF all, Strong, Middle and Weak) downsampled to the same size?**
>
> Yes (see Appendix E line 668).
>
> 7. **Does page 19 equation (5) \pi_{ref} refer to the SFT policy?**
>
> Pi_ref does refer to the SFT policy, we will clarify this in our updated paper.
>
> 8. **Can more data provide more improvements?**
>
> We first note that as seen in Table 1, datasets that are simply larger are not necessarily more performant - SHP-2 and StackExchange are by far the largest datasets we test, but underperform much smaller datasets. Similarly, Nectar is 3x larger than UltraFeedback but performs slightly worse for DPO. However, we do also observe some evidence that increasing dataset size (or at least, increasing up to 60k samples) helps: training on all UltraFeedback data outperforms training on the {weak,middle,strong} subsets (all of which consist of 10k datapoints) - see Table 7 in the Appendix.
>
> 9. **I am curious about the training process during these experiments.**
>
> We recorded logs of reward, KL divergences, average response lengths, and more for all our experiments. We observed that average model output length tended to increase over training, as observed in prior work. Interestingly, we also observed that performance on different specific evaluations behaved differently over training, and provide further details in the additional rebuttal material PDF - please see the PDF attached to the general response.
>
>
> [1] Cui et al. (2023). UltraFeedback: Boosting Language Models with Scaled AI Feedback. ICML.
>
> [2] Xu et al (2024). Is DPO Superior to PPO for LLM Alignment? A Comprehensive Study.
>
> [3] Tajwar, et al  (2024). Preference Fine-Tuning of LLMs Should Leverage Suboptimal, On-Policy Data.

---

> > ### Comment · Reviewer_WxZd · 2024-08-13
> >
> > Thank you for your response, I decide to keep my score.

---

> > > ### Author Response · Authors · 2024-08-13
> > >
> > > Thank you for responding! We are happy to clarify any further questions or address concerns if you have other reasons for not raising the score beyond the original weaknesses and questions asked (which we hope we have addressed appropriately above).

---

### Official Review · Reviewer_mnCL · 2024-07-10

**Soundness:** 2
**Presentation:** 3
**Contribution:** 3
**Rating:** 6
**Confidence:** 3

**Summary:**

This paper disentangles core components of current learning from preference feedback algorithms in alignment, conducts comprehensive experiments on the individual effect of each component, and provides a recipe of learning from preference feedback based on experiment results.

**Strengths:**

1. This paper aims to understand PPO and DPO from a practical perspective by conducting comprehensive experiments on core components of the RLHF pipeline. The results enhance the understanding of RLHF and provide valuable reference data for the community.
2. This paper is well-written and clearly presented.

**Weaknesses:**

1. The experiment testing the influence of training dataset size for the reward model is not sound enough. The dataset is a mixture of different high-quality datasets, which may affect the reward model's performance on different tasks, making it difficult to attribute changes in performance solely to the dataset size. Besides, the conclusion that "both increasing the reward model dataset (‘Mix’) and reward model size (from 13B to 70B) improve performance" is not evident, as the dataset size shows little influence in the 70B model.
2. The limited number of models tested in the paper may restrict the impact of this work, as results could vary among different models, as the authors have mentioned in their "Limitations" section.

**Questions:**

Here are some questions that I'd like to discuss with the authors to enhance my understanding of this work:
1. What is the data ratio of Mix and UltraF in the reward model experiment?
2. What is the codebase used in this paper?
3. On the right side of Table 3, the GSM scores indicate that the 70B UltraF.RM with UltraF.prompts performs best, whereas Figure 3 shows a different result. Is this discrepancy due to different experimental settings?

---

> ### Author Rebuttal · Authors · 2024-08-07
>
> Thank you very much for your review and feedback, and for noting that our results are a valuable reference for the community and enhance understanding of RLHF. We address your feedback and questions below:
>
> **Concerns**:
> 1. **Testing the influence of the training dataset size on reward model & downstream performance.**
>
> Our focus in these experiments is to see if improvements to the reward model (either through incorporating different data or increasing the RM size) lead to improvements in the downstream policy trained through PPO. In this sense, we are interested to see that both types of changes did not yield consistent large changes in downstream performance in Table 3, despite the fact that we observe improvements in RewardBench and Best-of-N settings, which more directly test RM performance.
> We will clarify that "both increasing the reward model dataset (‘Mix’) and reward model size (from 13B to 70B) improve performance" means that taking either option may improve performance, as indeed we agree that we do not see evidence that a larger dataset together with a larger RM further improves performance.
>
> 2. **Limited number of models tested.**
>
> Please see our general response, point 1. We provide additional results using Llama 3 and think further extending our observations to other models would be interesting future work.
>
> **Questions**:
> 1. **What is the data ratio of Mix and UltraF?**
>
> We provide exact sizes of the subsets and overall sizes of Mix and UltraF in Table 6 in the appendix. Mix contains roughly 260k samples from Stack Exchange, HH-RLHF, HelpSteer, PRM800k, and Nectar. UltraF contains roughly 60k samples. UltraFeedback itself is made up of prompts and completions (from varied models) from FalseQA, Evol Instruct, TruthfulQA, Flan V2, ShareGPT, and UltraChat (see Table 7 in Appendix E or the UltraFeedback paper itself [1] for more details).
> We add HelpSteer, Nectar, and HH-RLHF as they are the next best-performing datasets after UltraFeedback, and downsample HH-RLHF to avoid it making up most of the mixture. We then add Stack Exchange and PRM800k data to further diversify the data mixture, adding in more code and math-related data, which we empirically confirm aids reward model performance (see Table 3).
>
>
> 2. **What is the codebase used in this paper?**
>
> We extend EasyLM (https://github.com/young-geng/EasyLM) with our own DPO and PPO implementations to make the training work on our infrastructure, similar to Tulu 2. We compare our implementation details to InstructGPT, Rainier/Crystal, Finegrained-RLHF, AlpacaFarm, and Quark in Table 10 in Appendix F.2, and referenced the hyperparameters used in these approaches (especially InstructGPT, AlpacaFarm, and the Llama-2 paper) when choosing hyperparameters for our own experiments. We will release a link to the code after the anonymity period.
>
>
> 3. **Table 3 vs Figure 3 unmatched GSM scores**
>
> Yes, the different numbers reflect different experimental settings. In Figure 3, we limit training to 20k prompts due to the small number of GSM8k training prompts, and also to ensure that we are able to pick more relevant prompts for the mined set - we will clarify this in the figure caption.
>
> [1] Cui et al. (2023). UltraFeedback: Boosting Language Models with Scaled AI Feedback. ICML.

---

> > ### Comment · Reviewer_mnCL · 2024-08-11
> >
> > Thank you for your response, most of my concerns are addressed during rebuttal and I decide to keep my score.

---

### Author Rebuttal · Authors · 2024-08-07

We thank all the reviewers for their comments and feedback. We are happy that reviewers have noted our results and findings are of interest to the community (mnCL, WxZd, wF89), and enhances understanding of RLHF (mnCL), with comprehensive experiments/datasets (P8L3, mnCL). Additionally, we are happy that reviewers noted that we describe our experimental setup comprehensively (P8L3, wF89), and that the paper overall is clearly written and presented (mnCL, wF89, P8L3).

We address some common concerns below:

1. **We base our work on one pretrained model + More ablations & experiment combinations could be useful in further validating our hypotheses** (WxZd, mnCL, P8L3).

First, our end goal is to develop a recipe for strong open model performance, using publicly-available resources (hence our choice of Tulu 2, a state-of-the-art open LM with available SFT checkpoints). Focussing on 1 model allows us to explore other factors (such as dataset choice, RM choice, algorithm choice, etc) more cleanly and without larger computational costs. We aim to be comprehensive and varied in testing RLHF datasets (testing 12 varied datasets) and ablations of PPO itself (testing 4 different RMs each with 4 different prompt sets - GSM8k, mined prompts, mixed, and UltraFeedback, alongside the PPO experiments across varied datasets). Additionally, we note that prior work exploring RLHF methods often similarly focus on one model [1,2,3,4, inter alia]. We agree that further exploring how our results apply to other base models would be interesting, although it would require a large computational budget due to the number of ablations to run. Relatedly, our computational limitations meant we could not run every possible ablation (for example, we could not do many PPO training runs with a 70B RM, and so had to be selective in what to run there), and we note that reviewers generally agreed that the findings in our work are of interest to researchers in the field. However, we do additionally report results applying our recipe to Llama 3 8B, first finetuning Llama 3 on Tulu 2 and then performing DPO/PPO training with UltraFeedback and the hyperparameters given in our work (we use an 8B and not a 70B RM due to computational limits):

| Model | factuality | reason. | coding | inst. foll. | safety | truthf. | avg |
|- | - | - | - | - |- |- | - |
| Llama 3 + Tulu 2 SFT | 58.0 | 58.6 | 56.4 | 42.6 | 92.8 | 59.2 | 61.3 |
| +DPO	| 59.4 | 56.2 | 55.6 | 50.4 | 91.7 | 71.4 | 64.1 |
|+PPO	| 59.5 | 57.0 | 55.9 |56.0 | 91.4 | 69.6 | **64.9** |

Similarly to our prior results, we find better performance using PPO with UltraFeedback, and both approaches provide improvements over the SFT base model. Additionally, these models outperform our llama 2-based models on average. We believe further extending our work to other models, or to multimodal models (as pointed out by reviewer wF89). Finally, we additionally provide a PDF with this general response detailing an investigation looking at the performance of a 13B Tulu 2 model trained using PPO and a 70B UltraF. RM with UltraF. prompts over the course of training, showing how different evaluations are affected by our PPO training.

2. **Ensuring sound experimental setup and matching claims** (P8L3).

We made best-effort attempts to tune both PPO and DPO and compare them in a fair manner, ensuring that we had solid hyperparameters for both without doing a grid search for every experiment (important as PPO experiments took ~ 3 days to run on our compute). We performed much initial tuning of both PPO and DPO in initial experiments on HH-RLHF and UltraFeedback, and provide details in our response to reviewer P8L3 (concern 1).

Additionally, we clarify that our claims are made in the context of our empirical work, and will make this context more explicit in the text. Our intention is to explore the performance of two popular RLHF methods using public data (within our computational budget), and so our claims hold strictly only for the datasets and models tested, although we did our best to cover as many datasets as feasible with our computational budget (13 for DPO and 5 for PPO), as well as further PPO settings (4 different reward models up to 70B in size). We will clarify our claims where reviewers have pointed out they may be unclear.

---
[1] Sun et al. (2024). SALMON: Self-Alignment with Instructable Reward Models. ICLR.

[2] Shen et al. (2024). The Trickle-down Impact of Reward (In-)consistency on RLHF. ICLR.

[3] Wu et al. (2023). Fine-Grained Human Feedback Gives Better Rewards for Language Model Training. NeurIPS.

[4] Cui et al. (2024). UltraFeedback: Boosting Language Models with Scaled AI Feedback. ICML.

---

### Decision · Program_Chairs · 2024-09-25

**Decision:**

Accept (poster)

**Comment:**

The primary concerns raised by the reviewers related to the limitations in the breadth of the experiments and the depth of analysis. However, these concerns were largely addressed by the additional experiments during the rebuttal period.  The authors have argued that the constraints were due to computational limitations, which were managed effectively to extract meaningful insights from the experiments conducted. Moreover, the empirical results obtained are still highly valuable. The observations made in this paper are practical and provide a good foundation for further research. We recommend the paper for acceptance.